# Effect of Aromatic Herbs and Spices Present in the Mediterranean Diet on the Glycemic Profile in Type 2 Diabetes Subjects: A Systematic Review and Meta-Analysis

**DOI:** 10.3390/nu16060756

**Published:** 2024-03-07

**Authors:** María Carmen Garza, Sofía Pérez-Calahorra, Carmen Rodrigo-Carbó, María Antonia Sánchez-Calavera, Estíbaliz Jarauta, Rocío Mateo-Gallego, Irene Gracia-Rubio, Itziar Lamiquiz-Moneo

**Affiliations:** 1Department of Human Anatomy and Histology, School Medicine, University of Zaragoza, 50009 Zaragoza, Spain; mcgarza@unizar.es (M.C.G.); irenegraciarubio@gmail.com (I.G.-R.); itziarlamiquiz@unizar.es (I.L.-M.); 2Department of Physiatry and Nursing, Faculty of Health Science, University of Zaragoza, 50009 Zaragoza, Spain; spperezc@unizar.es; 3Unidad Clínica y de Investigación en Lípidos y Arteriosclerosis, Hospital Universitario Miguel Servet, Instituto de Investigación Sanitaria Aragón (IIS Aragón), Centro de Investigación Biomédica en Red Enfermedades Cardiovasculares (CIBERCV), 50009 Zaragoza, Spain; crodrigocarbo@gmail.com (C.R.-C.); estijarauta@gmail.com (E.J.); 4Department of Medicine and Psychiatry, University of Zaragoza, 50009 Zaragoza, Spain; 5Health Research Institute of Aragon (IIS Aragón), 50009 Zaragoza, Spain; 6Aragonés Health Service, 50009 Zaragoza, Spain; 7Research Network on Preventive Activities and Health Promotion (Red de Investigación en Actividades Preventivas y Promoción de la Salud), 08007 Barcelona, Spain

**Keywords:** Mediterranean Diet, Type 2 Diabetes, aromatic herbs, spices

## Abstract

Background: The Mediterranean Diet (MedDiet) is the dietary pattern par excellence for managing and preventing metabolic diseases, such as Type 2 Diabetes (T2DM). The MedDiet incorporates spices and aromatic herbs, which are abundant sources of bioactive compounds. The aim of this study was to analyze the effect of all aromatic herbs and spices included in the MedDiet, such as black cumin, clove, parsley, saffron, thyme, ginger, black pepper, rosemary, turmeric, basil, oregano, and cinnamon, on the glycemic profile in T2DM subjects. Methods: PubMed, Web of Science, and Scopus databases were searched for interventional studies investigating the effect of these aromatic herbs and spices on the glycemic profile in T2DM subjects. Results: This systematic review retrieved 6958 studies, of which 77 were included in the qualitative synthesis and 45 were included in the meta-analysis. Our results showed that cinnamon, turmeric, ginger, black cumin, and saffron significantly improved the fasting glucose levels in T2DM subjects. The most significant decreases in fasting glucose were achieved after supplementation with black cumin, followed by cinnamon and ginger, which achieved a decrease of between 27 and 17 mg/dL. Conclusions: Only ginger and black cumin reported a significant improvement in glycated hemoglobin, and only cinnamon and ginger showed a significant decrease in insulin.

## 1. Introduction

Diabetes Mellitus (DM) and, specifically, Type 2 Diabetes (T2DM) have emerged as an increasingly critical healthcare priority. Over the past four decades, the number of people affected by DM has dramatically risen, exceeding 460 million individuals today [1]. Ten years after diagnosis, approximately 60% of patients are estimated to have three or more comorbidities, directly contributing to 6.7 million deaths each year [2]. T2DM is characterized by varying degrees of insulin resistance and beta cell dysfunction, with its development influenced by a range of risk factors, including genetic, metabolic, and environmental factors [3]. Although individual predisposition to T2DM is substantially shaped by non-modifiable risk factors such as ethnicity and family history/genetic predisposition, epidemiological studies highlight the potential for preventing a significant number of T2DM cases by improving critical modifiable risk factors, such as obesity, physical inactivity, and an unhealthy diet [4,5,6]. Therefore, dietary guidance is crucial for enhancing both lifespan and overall quality of life in T2DM patients [7]. 

The Mediterranean Diet (MedDiet) reflects the traditional dietary pattern observed in regions where olive trees are cultivated, such as Crete, Greece, and Southern Italy. This diet is characterized by a substantial intake of fats, primarily in the form of extra-virgin olive oil. It also involves a high consumption of low-glycemic-index carbohydrate-rich foods such as whole-grain cereals, legumes, nuts, fruits, and vegetables. Additionally, it includes a moderate-to-high consumption of fish, poultry, and dairy products in moderate-to-small quantities. Red meat and meat products are limited, and there is a moderate intake of alcohol, primarily in the form of red wine [8,9]. The PREDIMED study, encompassing 7447 participants, employed a randomized design with three dietary groups. One group followed the MedDiet supplemented with extra-virgin olive oil, another group adhered to the MedDiet supplemented with mixed nuts, and the control group received advice on a low-fat diet [10]. This study demonstrated that the MedDiet had a positive impact on two prevalent conditions strongly linked to adiposity: metabolic syndrome [11] and T2DM [12]. The MedDiet not only lowered the risk of diabetes among individuals with high cardiovascular risk [12] but also improved the glycemic profile in T2DM subjects [13]. The MedDiet promotes incorporating spices, aromatic herbs, garlic, and onion to introduce a diverse range of flavors and enhance the palatability of dishes. This approach also provides an opportunity to reduce the use of salt, which is a significant contributor to the development of hypertension in predisposed individuals [8]. Furthermore, culinary aromatic herbs and spices are abundant sources of bioactive compounds, including sulfur-containing substances, tannins, alkaloids, phenolic diterpenes, and vitamins, particularly flavonoids and polyphenols [14,15]. These bioactive compounds could exhibit antioxidative, anti-inflammatory, antitumor, anticarcinogenic, and blood-sugar- and cholesterol-lowering properties [16]. Therefore, the aim of this study was to analyze the effect of all aromatic herbs and spices included in the MedDiet, such as black cumin, clove, parsley, saffron, thyme, ginger, black pepper, rosemary, turmeric, basil, oregano, and cinnamon, on the glycemic profile in T2DM subjects.

## 2. Materials and Methods

This meta-analysis has been reported according to the Preferred Reporting Items for Systematic Reviews and Meta-Analyses (PRISMA) guidelines [17]. The PRISMA checklist is available in Appendix A.

### 2.1. Search Strategy and Study Selection

A systematic search of the relevant literature was performed using three citation databases, including PubMed, Web of Science, and Scopus, in order to identify interventional studies investigating the effect of different aromatic herb supplementation, commonly used in the Mediterranean Diet, on the glucose profile in T2DM subjects. Articles cited in reviews addressing this topic were checked and included in this study if necessary. The search strategy involved the terms for the aromatic herbs and spices studied and for the outcomes related to glycemic profile, obtaining the following search combinations: [(NIGELLA SATIVA [Title/Abstract]) OR (BLACK CUMIN[Title/Abstract]); (SYZYGIUM AROMATICUM [Title/Abstract]) OR (CLOVE[Title/Abstract]); (PETROSELINUM CRISPUM [Title/Abstract]) OR (PARSLEY [Title/Abstract]); (CROCUS SATIVUS [Title/Abstract]) OR (SAFFRON [Title/Abstract]); (THYMUS VULGARIS [Title/Abstract]) OR (THYME [Title/Abstract]); (ZINGIBER OFFICINALE [Title/Abstract]) OR (GINGER [Title/Abstract]); (PIPER NIGRUM [Title/Abstract]) OR (BLACK PEPPER [Title/Abstract]); (SALVIA ROSMARINUS [Title/Abstract]) OR (ROSEMARY [Title/Abstract]); (CURCUMA LONGA [Title/Abstract]) OR (TURMERIC [Title/Abstract]); AND (DIABETES[Title/Abstract]) OR (GLUCOSE[Title/Abstract]) OR (INSULIN[Title/Abstract])]. 

Articles retrieved until September 2023 were then included or excluded based on the following criteria. The inclusion criteria involved (a) articles published in a peer-reviewed journal; (b) and interventional studies; (c) studies conducted in adults; (d) studies conducted in humans with T2DM; (e) studies which included any supplementation with black cumin, clove, parsley, saffron, thyme, ginger, black pepper, rosemary, curcumin, cinnamon, basil, and/or oregano; and (f) studies which reported data about fasting glucose and/or glycated hemoglobin (HbA1c) and/or insulin. The exclusion criteria included (a) case studies; (b) letters, commentaries, conference papers, and narrative reviews; (c) studies not conducted in humans; and (d) studies conducted in children. The search was limited to the literature presented in English. 

### 2.2. Outcome Measures

The primary outcomes of interest were changes in fasting glucose, insulin, and HbA1c. Body weight and body mass index (BMI) variation after intervention were secondary outcomes. 

### 2.3. Data Collection and Data Synthesis

Glucose metabolism and body weight outcomes were extracted and recorded in a database for analysis. This included mean values before and after intervention, alongside standard deviations. If not explicitly stated, the difference between pre-intervention and post-intervention means was calculated by subtracting the baseline from post-intervention values. This difference was derived as a change from the baseline and applied consistently when different methods were used to measure the same outcome. The standard deviation of the mean difference was computed as follows: SD = square root [(SD pre-intervention)^2^ + (SD post-intervention)^2^ − (2R × SD pre-intervention × SD post-intervention)], assuming an effect model due to a moderate level of heterogeneity (>50%), which was quantitatively assessed using the Higgins index *I*^2^. If necessary, authors of the studies included were contacted to acquire missing values.

### 2.4. Statistical Analysis

Statistical analysis was performed using statistical computing with a package (meta) in R software (version 3.5.0) [18], as was previously reported by Mateo-Gallego et al. [19]. Briefly, between-group meta-analyses were completed for continuous data using change in mean and standard deviation. Heterogeneity was analyzed using Cochrane Q and Higgins *I*^2^ tests, and Egger plots were used to assess the risk of publication bias (Appendix A). The level of significance was set at *p* < 0.05 and with 95% confidence intervals.

### 2.5. Quality Measures

The quality of each included trial was assessed based on the previously validated methodology developed by Kmet et al. [20]. The procedure was derived from a checklist for determining the quality of quantitative studies, which included fourteen questions previously described [19,21]. Each question can be answered with “yes”, “partial”, “no”, or “not applicable”. Scoring followed the following formula: ((number of “yes” × 2) + (number of “partial” × 1))/(total possible sum (28) − (number of “not applicable” × 2)). Scores ranged from 0 to 1, with higher values indicating higher trial quality. The quality assessment of each trial involved three researchers (ILM, MCG, and SPC). Two researchers conducted the trial’s quality checklist; if there was a discrepancy (a mean score difference of more than 0.1 points), the third researcher conducted an additional review to resolve it.

## 3. Results

### 3.1. Study Selection

The systematic search retrieved 6958 studies of which 2641 were identified in PubMed, 1152 in Web of Science, and 3165 in Scopus. After removing duplicated articles (*n* = 2137), 4821 manuscripts were screened, excluding 2077 for not being carried out in humans or not being clinical trials. The abstracts of the remaining 2564 articles were reviewed, leading to the exclusion of 2299 articles for not meeting the selection criteria. A full-text review was then conducted on 265 articles, with 188 being excluded for various reasons: no individuals with T2DM (*n* = 68), no reporting fasting glycemic metabolism parameters (*n* = 58), in vitro results (*n* = 17), reused data (*n* = 35), and letters to the editor (*n* = 10). Finally, seventy-seven articles fulfilled the eligibility criteria to be included in the systematic review, and out of those articles, 45 were included in the quantitative synthesis (meta-analysis). The reasons for excluding 32 studies from the qualitative synthesis were as follows: failure to use a control group or use of an inappropriate control that received an antidiabetic drug (*n* = 17), insufficient data (*n* = 8), and use of mixed herbs (*n* = 7). Of the 77 articles included in this systematic review, the analyzed herbs were cinnamon, curcumin, ginger, black cumin, saffron, and rosemary. All studies including placebo and interventional groups, and displaying glycemic profile values pre- and post-supplementation, were included in the quantitative synthesis, obtaining a total of forty-five articles from those seventy-seven: ten examined the effect of saffron supplementation, eight examined the effect of black cumin, nine examined the effect of ginger, seven analyzed the effect of curcumin, ten examined the effect of cinnamon, and one study analyzed the effect of cinnamon, cardamon, saffron, and ginger with a five-arm study (Figure 1).

### 3.2. Participants and Main Study Characteristics

A detailed description of the studies included in the meta-analysis can be found in Table 1. The 45 studies gathered information on a total of 3050 participants (aged 18–80 years). There was some heterogeneity in the clinical characteristics of the study populations. In summary, twenty studies recruited non-insulin-dependent T2DM subjects with the following characteristics [22,23,24,25,26,27,28,29,30,31,32,33,34,35,36,37,38,39,40,41]: in nine studies, the subjects received only oral antidiabetic drugs [41,42,43,44,45,46,47,48,49]; in three studies, the subjects were newly T2DM-diagnosed subjects [50,51,52]; in three studies, the T2DM subjects had a chronic renal disease [53,54,55]; in two studies, the T2DM subjects reported levels of HbA1c higher than 7% [56,57]; in two studies, the T2DM subjects had normal blood pressure [58,59]; in two studies, the T2DM subjects were women without cardiovascular disease [60,61]; in one study, the subjects were T2DM postmenopausal women [62]; in one study, the T2DM subjects also had hyperlipidemia [63]; and in one study, T2DM subjects also had a metabolic syndrome [64]. One of them included T2DM subjects taking stable T2DM medications for two months [65].

Regarding the population gender, 37 out of the 45 studies recruited participants of both sexes. In the remaining studies, three included only women [60,61,62], one recruited just T2DM men [51], and four did not indicate the sex of participants [31,34,44,55]. Of the forty-five articles included in the meta-analysis, ten analyzed the effect of cinnamon on T2DM subjects, seven had a two-arm intervention [22,23,25,43,50,57,62], two had a three-arm intervention [24,56], and one had a six-arm intervention [26]. These studies administered varying dosages of cinnamon, ranging from 360 to 3000 mg, with 1000 mg being the most commonly used. Seven studies analyzed the effect of turmeric on T2DM subjects, with five employing a two-arm intervention [27,28,51,57,63], one using a three-arm intervention [65], and one opting for a four-arm intervention [34]. The dosage of turmeric supplementation varied between 80 and 2000 mg, with 2000 mg being the most prevalent. Nine studies analyzed the effect of ginger in T2DM subjects, all of which utilized a two-arm intervention [29,30,31,35,36,49,52,53,59]. However, there was considerable heterogeneity in the dosage of ginger supplementation, ranging from 600 to 3000 mg, with 2000 mg being the most frequently employed. Eight studies analyzed the effect of black cumin in T2DM subjects, with seven employing two-arm intervention [37,38,46,47,54,55,64], while only one utilized a four-arm intervention [60]. The dosage of black cumin supplementation ranged from 500 mg to 3000 mg, although 500 mg was the most commonly administered dosage. Ten studies analyzed the effect of saffron in T2DM subjects, with eight utilizing a two-arm intervention [33,40,41,42,44,45,48,66], one employing a three-arm intervention [39], and another opting for a four-arm intervention [61]. Saffron supplementation dosage ranged from 15 to 3 g, with the most common dosage ranging between 30 and 100 mg. Finally, one article examined the effect of cinnamon, cardamom, saffron, and ginger on T2DM subjects, including a four-arm intervention, one of each spice, without a placebo or control group [32] (Table 1).

Appendix A shows the main characteristics of the 32 articles included in the qualitative analysis. The studies included a total of 2398 participants with an age range of between 30 and 70 years. Thirty studies recruited subjects with T2DM [67,68,69,70,71,72,73,74,75,76,77,78,79,80,81,82,83,84,85,86,87,88,89,90,91,92,93,94,95,96], one study included prediabetic and newly diagnosed diabetic subjects [97], and another study involved participants diagnosed with T2DM alongside obese individuals [98]. Eight studies reported a single-arm intervention in T2DM subjects, one supplemented with rosemary [70], three analyzed the effect of black cumin [74,76,98] on the glycemic profile, three supplemented with curcumin [77,81,82], and one examined the effect of ginger on the glycemic profile [89]. Thirteen studies utilized a two-arm intervention in T2DM subjects; two analyzed the effect of black cumin [71,73], five supplemented with turmeric [83,84,86,87,88], two analyzed the effect of ginger [67,90], and four supplemented with cinnamon [91,93,94,95]. Six studies employed a three-arm intervention: one supplemented with cinnamon [69], two analyzed the effect of black cumin [72,75], two supplemented with curcumin [79,85], and another one supplemented with a mix of herbs [80]. Four studies had a four-arm intervention, which included the effect of cinnamon supplementation [93,96], one analyzed the effect of saffron [68], and another one used a mix of herbs and spices [97], and only one had a six-arm interventions, which included a herbal mix containing turmeric [78].

### 3.3. Aromatic Herb Supplementation

Regarding cinnamon supplementation, the ten studies included in the meta-analysis used capsules to achieve the supplementation, although with very heterogeneous dosages, and one study used cinnamon included in black tea. Most studies prescribed 1000 mg of cinnamon per day [23,24,50,57]. However, some reach 1500 [62] or even 3000 mg per day [25,26], while another one prescribed only between 120 to 360 mg per day [56].

All seven articles that investigated the impact of curcumin on the glycemic profile utilized capsules for supplementation, with widely varying dosages. Three studies prescribed 2000–2100 mg of curcumin per day [34,51,63]; however, others prescribed less, reaching 1500 mg per day [58], while others prescribe much lower dosages, with only 500 mg per day [28] or 150 mg per day [65] or even 80 mg per day [27].

Out of the ten articles analyzing the effect of ginger on the glycemic profile, nine used capsules with very heterogeneous dosages, and only one included ginger supplementation in black tea. Most studies prescribed 2000 mg of ginger per day [29,36,53,59], although others prescribed more, reaching 3000 mg/day [30,31], while others prescribed lower dosages, 1600 mg per day [35] or even less with only 600 mg per day [49,52].

Regarding black cumin supplementation, six out of eight studies included in this meta-analysis used capsules to achieve the supplementation [37,38,47,54,64], while the other two studies used oils [46,55]. Black cumin supplementation included heterogeneous dosages, including 500 mg per day [37,64], 1000 mg per day [38], 2000 mg per day [54,60], and 3000 mg per day [47]. Among the two studies that prescribed black cumin oil, one provided 5 mL per day [46], while the other one provided 2.5 mL per day [55].

Out of the eleven studies analyzing the effect of saffron on the glycemic profile, ten administered saffron in capsule form, with varying dosages, while one study supplemented saffron with black tea. The study with the highest dosage involved a four-arm approach, with a supplementation of 3 g of ginger, 3 g of cardamom, 3 g of cinnamon, and 3 g of saffron in black tea [32], followed by one study prescribing 400 mg per day [61] and other two studies prescribing 100 mg per day [33,40,44]. Studies prescribing lower dosages administered 30 mg per day [41,42,48] or 15 mg per day [39,45,66].

### 3.4. Changes in Glycemic Metabolism

#### 3.4.1. Fasting Glucose

Ten out of eleven studies prescribing cinnamon included in the current meta-analysis reported fasting glucose data pre- and post-supplementation, and six observed significant differences after intervention (Table 1). Akilen et al. [43], Davari et al. [50], and Talaei et al. [25] showed a slight but not significant decrease in the cinnamon supplementation arm, while those participants receiving a placebo showed a slight increase in fasting glucose concentrations. Vanschoonbeek et al. [62] also showed a slight decrease in both the placebo group and the group supplemented with cinnamon but with non-significant differences after the intervention. In contrast, Lira Neto et al. [22], Mang et al. [23], Mirfeizi et al. [24], Khan et al. [26], Azimi et al. [32], and Lu et al. [56] found that fasting glucose only decreased in the group supplemented with cinnamon after the intervention. Especially notable is the case of Khan et al., who found a significant decrease in fasting glucose regardless of the dosage of cinnamon provided, ranging from 1 to 3 g per day. Data on fasting glucose reported by ten studies were included in the meta-analysis, all of which compared cinnamon supplementation vs. placebo supplementation. There was a reduction in fasting glucose of 18.67 mg/dL (−27.24 to −10.10 mg/dL, *p* < 0.001, Figure 2A) in the cinnamon supplementation group versus the placebo group. However, this reduction was not significantly different, including the predictive value (−46.84 to 9.50 mg/dL, Figure 2A).

All seven studies which prescribed curcumin in the current meta-analysis reported fasting glucose data pre- and post-supplementation, but only four described significant differences throughout the intervention. Of the three studies that did not find significant changes in fasting glucose after supplementation, two showed slight decreases pre- and post-intervention [63,65], while another one [58] showed a slight increase in fasting glucose concentration. In contrast, Asadi et al. [27] and Hodaei et al. [28] showed a significant decrease in fasting glucose after the curcumin supplementation. Selvi et al. showed that the greater the decrease in fasting glucose, the greater the dosage of turmeric supplied in [51]. At the same time, Darmian et al. [34] demonstrated that the decrease in fasting glucose was more significant when it was combined with physical activity. Data on fasting glucose reported by seven studies were included in the meta-analysis, and all of them compared curcumin versus placebo supplementation. There was a reduction in fasting glucose of 12.55 mg/dL (−14.18 to −10.86 mg/dL, *p* < 0.001, Figure 3A) in the curcumin versus the placebo group. This reduction was significantly different, including the predictive value (−14.10 to −10.34 mg/dL, Figure 3A).

Out of the ten articles analyzing the effect of ginger on the glycemic profile, six reported a significant decrease in the fasting glucose pre- and post-supplementation, while four did not find significant differences after the intervention. Arablou et al. [35], Arzati et al. [36], and Azim et al. [32] reported a slight not significant decrease in fasting glucose after supplementation, while Mahluji et al. [59] showed a slight increase in both the supplementation and placebo groups. Conversely, four studies reported a significant decrease only in the supplemented group at the end of the intervention [29,30,31,53]. Another two studies showed that the decrease in fasting glucose occurred both in the group that received ginger and the placebo group [49,52]. Our meta-analysis shows that there was a reduction in fasting glucose of 17.12 mg/dL (−29.60 to −4.64 mg/dL, *p* = 0.0004, Figure 4A) in the ginger supplementation versus the placebo group. However, this reduction was not significantly different, including the predictive value (−56.61 to 22.36 mg/dL, Figure 4A).

Among eight studies which prescribed black cumin as a supplement and were included in the meta-analysis, all reported significant differences in fasting glucose pre- and post-supplementation. Six showed a significant decrease in fasting glucose only in the supplemented group [37,38,46,47,54,64], while Ansari et al. [55] reported a significant decrease both in the black cumin and placebo groups [55]. In the same line, Jangjo-Borazjani et al. [60] reported a significant decrease in fasting glucose in both groups receiving either only black cumin supplementation or in combination with physical exercise. Our meta-analysis shows a reduction in fasting glucose of 26.33 mg/dL (−39.89 to −12.77 mg/dL, *p* = 0.0001, Figure 5A) in the black cumin supplementation group versus the placebo group. However, this reduction was not significantly different, including the predictive value (−71.46 to 18.80 mg/dL, Figure 5A).

Of the eleven articles analyzing the effect of saffron on the glycemic profile, six reported a substantial decrease in the fasting glucose pre- and post-supplementation [33,41,42,44,45,48]. Three trials did not find significant differences after intervention with saffron [32,40,66]. Sepahi et al. reported a significant decrease in those participants supplemented with crocin (a constituent of saffron) but not in those advised to take saffron [39]. On the other hand, Rajabi et al. [61] found a significant reduction in fasting glucose when combining saffron with physical exercise but not in the group receiving supplementation alone [61]. Our meta-analysis shows a reduction in fasting glucose of 7.06 mg/dL (−13.01 to −1.10 mg/dL, *p* = 0.020, Figure 6A) in the saffron supplementation versus placebo group. However, this reduction was not significantly different, including the predictive value (−22.09 to 7.98 mg/dL, Figure 6A).

#### 3.4.2. HbA1c

Among the eleven studies which supplemented with cinnamon, ten reported HbA1c pre- and post-intervention. Only four showed a significant decrease in HbA1c after cinnamon supplementation [24,43,56,57]. Data on HbA1c reported by ten studies were included in the meta-analysis, revealing a non-significant reduction in HbA1c of 0.04% (−0.08 to 0.00%, *p* = 0.0693, Figure 2B) in the cinnamon supplementation versus the placebo group.

Out of the seven studies analyzing the effect of curcumin supplementation, all of them reported HbA1c pre- and post-supplementation, with only three of them showing a significant decrease after curcumin supplementation [27,34,51]. The meta-analysis showed a non-significant reduction in HbA1c of 0.22% (−0.59 to 0.15%, *p* = 0.2370, Figure 3B) in the curcumin supplementation versus placebo group. Visual interpretation of funnel and bubble plots suggested limited publication bias in HbA1c levels comparing curcumin versus placebo supplementation (*p* = 0.0421, Appendix A).

Among the ten studies which analyzed the effect of ginger supplementation on the glycemic profile, nine of them reported HbA1c pre- and post-supplementation, of which five reported a significant decrease in HbA1c [29,30,31,35,52]. Data on HbA1c reported by nine studies were included in the meta-analysis, and all of them compared ginger versus placebo supplementation. The meta-analysis showed a significant reduction in HbA1c of 0.56% (−0.90 to −0.22%, *p* = 0.0013, Figure 4B) in the ginger supplementation versus placebo group.

Among the eight studies that analyzed the impact of black cumin supplementation on the glycemic profile, five of them reported HbA1c pre- and post-supplementation. All of them showed a significant decrease in the HbA1c after black cumin supplementation [37,46,47,54,64], regardless of whether the administration system was capsule or oil. Data on HbA1c reported by five studies were included in the meta-analysis, and all of them compared black cumin vs. placebo supplementation. The meta-analysis showed a significant reduction in HbA1c of 0.41% (−0.81 to −0.02%, *p* = 0. 0.0409, Figure 5B) in the black cumin supplementation versus the placebo group.

Eleven studies have analyzed the effect of saffron supplementation on the glycemic profile, of which eight reported HbA1c values pre- and post-supplementation. Only four of them reported a significant decrease in the HbA1c after saffron supplementation [39,40,42,48]. Data on HbA1c reported by eight studies were included in the meta-analysis, and all of them compared saffron versus placebo supplementation. The meta-analysis showed a non-significant reduction in HbA1c of 0.20% (−0.43 to 0.03%, *p* = 0.0941, Figure 6B) in the cinnamon supplementation versus the placebo group. Visual interpretation of funnel and bubble plots suggested limited publication bias in HbA1c levels comparing saffron versus placebo supplementation (*p* < 0.0001, Appendix A).

#### 3.4.3. Insulin

Eleven studies analyzed the effect of cinnamon supplementation on the glycemic profile, and six of them reported insulin values pre- and post-supplementation, with only one of them showing a significant decrease in insulin levels [24]. Only three studies reported the mean and standard deviation of insulin pre- and post-supplementation and were included in the meta-analysis. They showed a significant reduction in insulin of 0.76 UI/µL (−1.13 to −0.39, *p* < 0.0001, Figure 7A) in the cinnamon supplementation versus placebo group.

Among the seven studies which analyzed the effect of curcumin on the glycemic profile, four reported insulin values pre- and post-supplementation, and only one showed a significant decrease [34]. Data on insulin reported by four studies were included in the meta-analysis, showing that there was a non-significant reduction in insulin of 2.36 UI/µL (−5.19 to 0.38 UI/µL, *p* = 0.0911, Figure 7B) in the curcumin supplementation versus the placebo group. Visual interpretation of funnel and bubble plots suggested limited publication bias in insulin levels comparing curcumin versus placebo supplementation (*p* = 0.0008, Appendix A).

Ten studies analyzed the effect of ginger on the glycemic profile, and six of them reported insulin values pre- and post-supplementation, with four of them showing a significant decrease [31,35,52,59]. Data on insulin reported by these six studies were included in the meta-analysis, showing a significant reduction in insulin of 1.69 UI/µL (−2.66 to 0.72 UI/µL, *p* = 0.0006, Figure 8A) in the ginger supplementation versus the placebo group.

Among the eight studies that analyzed the effect of black cumin on the glycemic profile, four reported insulin values pre- and post-supplementation, and two showed a significant decrease [54,60]. Data on insulin reported by these four studies were included in the meta-analysis, showing a non-significant increase in insulin of 1.68 UI/µL (−2.15 to 5.52 UI/µL, *p* = 0.3900, Figure 8B) in the black cumin supplementation versus the placebo group. Visual interpretation of funnel and bubble plots suggested limited publication bias in insulin levels comparing black cumin versus placebo supplementation (*p* = 0.0377, Appendix A).

Eleven studies analyzed the effect of saffron supplementation on the glycemic profile, with seven of them reporting insulin values pre- and post-supplementation. Among these, four studies demonstrated a significant decrease [33,39,45,61]. Data on insulin reported by these seven studies were included in the meta-analysis, showing that there was a non-significant decrease in insulin of 0.14 UI/µL (−1.94 to 1.67 UI/µL, *p* = 0.8809, Figure 9) in the saffron supplementation versus placebo group.

The overall quality score of the included studies in the meta-analysis is summarized in Table 2, with a quality score ranging from 0.36 to 0.95 and a mean score of 0.68. A detailed description of the quality assessment for each study is included in Table 2. The most outstanding concerning issues were the blinding of investigators and subjects, analytic methods, and controlling for confounding factors. Among the forty-five studies that were included in the meta-analysis, only four had a control for confounding, although most of them partially achieved it. In addition, only fourteen studies reported investigators’ blinding, and nine partially described it.

The overall quality score of the included studies in the review analysis is summarized in Appendix A. These studies showed lower quality scores than studies included in the meta-analysis, with a score that ranged from 0.25 to 0.93 and a mean score of 0.54. Appendix A shows the detailed description of the quality assessment for each study included in the systematic review. The greatest concerning issues were the blinding of investigators, sample size calculation, analytical methods, and controlling for confounding factors. In this regard, of the 32 studies included in the systematic review, only four carried out a blinded intervention, only two correctly used the analytical methods, only eleven calculated the sample size properly, and only two conducted a statistical analysis considering the confounding factors.

## 4. Discussion

As far as we are aware, this is the first systematic review and meta-analysis aiming to evaluate the effect of aromatic herbs and spices included in the MedDiet, such as black cumin, clove, parsley, saffron, thyme, ginger, black pepper, rosemary, turmeric, basil, oregano, and cinnamon, on the glycemic profile of individuals with T2DM. To develop this analysis, 77 articles fulfilled the eligibility criteria, of which 45 were included in the quantitative synthesis (meta-analysis) and 32 in the systematic review. Finally, only five out of the twelve aromatic herbs and spices were investigated; for the remaining ones (clove, parsley, thyme, black pepper, rosemary, basil, and oregano), not enough studies were found on the glycemic profile in T2DM subjects. Our results showed that cinnamon, turmeric, ginger, black cumin, and saffron significantly improved fasting glucose in T2DM subjects. However, the greatest decreases in fasting glucose, between 17 and 27 mg/dL, were achieved after supplementation with black cumin, followed by cinnamon and ginger. On the other hand, only ginger and black cumin reported a significant improvement in HbA1c, and only cinnamon and ginger showed a significant decrease in insulin values. According to the American Diabetes Association, fasting glucose and HbA1c are the reference parameters in the diagnosis and management of diabetic patients; meanwhile, the HbA1c is considered a value with more pre-analytical stability, i.e., less disturbance due to stress situations or changes in nutrition [99]. Hence, when focusing on HbA1c, only ginger and black cumin demonstrated therapeutic effects. However, our meta-analysis highlights ginger as a herb with substantial translational potential for diabetes treatment, impacting all three glycemic parameters. Regarding clove, parsley, thyme, black pepper, rosemary, basil, and oregano, more studies are needed to analyze the effect of these herbs on the glycemic profile in T2DM subjects.

Among the eleven studies that incorporated cinnamon in the current meta-analysis, six reported significant differences in fasting glucose [22,23,24,26,32,56] and four in the HbA1c [24,43,56,57] after the supplementation, whereas one showed a significant decrease in insulin levels [24]. The variation in study outcomes regarding the impact of cinnamon consumption on glycemic markers can be attributed to variations in several influential factors, including the use of concurrent medications, baseline fasting glucose levels, intervention duration, cinnamon dosage, ethnic background, and the BMI of the study participants [24,25]. In this sense, two studies [26,56] selected subjects using only sulfonylurea derivatives, another six studies [22,23,24,43,57,62] carried out the study in a cohort of patients who were prescribed commonly used combinations of oral blood-glucose-lowering medications, and another two studies chose participants exclusively on metformin treatment [25,50]. The intervention duration also has large variations, with a range from 40 days [26,62] to 112 days [23], and the cinnamon concentration varied from 120 mg [56] to 6 g [26] per day. It is worth mentioning that Lu et al. [56] observed significant differences in HbA1c and fasting glucose with the lowest concentration of cinnamon supplementation. In contrast, Davari et al. [50] and Talaei et al. [25] did not find significant differences with 3 g of cinnamon supplement. This discrepancy could be due to the fact that all patients in the trial conducted by Lu et al. [56] were taking the same type of prescribed antidiabetic medication. Our meta-analysis is the largest one, including eleven studies, and it revealed that subjects with T2DM who were supplemented with cinnamon obtained significant reductions in fasting glucose, greater than 18 mg/dL, and insulin levels compared with the placebo group. Several studies have shown that the bioactive extracts of cinnamon activated glycogen synthase, increased glucose uptake, and inhibited glycogen synthase kinase-3β [100,101]. Furthermore, sections of cinnamon also activated insulin receptor kinase and inhibited dephosphorylation of the insulin receptor 1 [101]. Indeed, these combined effects contribute to enhanced insulin sensitivity. They may serve as the mechanism underlying cinnamon’s influence on glycemic profiles [26].

In the current meta-analysis, seven studies analyzed the effect of curcumin supplementation, with four showing a significant difference in fasting glucose levels [27,28,34,51], three of them showing a significant difference in HbA1c [27,34,51], and only one of them showing a significant decrease in insulin levels [34]. The divergent results in the glycemic profile in the different studies could be attributed to variations in the utilization of whole turmeric powder versus curcumin, the bioactive polyphenol compound [102], treatment dosage, differences in study methodologies, and duration [63]. In this context, three studies [34,51,63] administered whole turmeric at similar dosages, and two of them observed significant differences in glucose parameters [34,51]. These discrepancies may be attributed to the fact that one of the studies administered turmeric in combination with metformin [51], while another incorporated physical exercise [34]. Consequently, the combined influence of these factors could potentially enhance the effects of turmeric. Curcumin was administrated in four studies [27,28,58,65] and improved the glycemic profile in two of them [28,57]. The absence of an enhancement in glycemic parameters could be due to T2DM patients enrolled in the study of Vanaie et al. [58], which included insulin-dependent individuals, as well as the relatively low dosage of curcumin supplementation in the study developed by Usharani et al. [65] (300 mg/day or 600 mg/day) compared to the dosage of curcumin administered in the study carried out by Hodaei et al. [28] (1500 mg/day). In contrast, Asadi et al. found a significant difference in fasting glucose levels and HbA1c in T2DM subjects supplemented with 80 mg/day of curcumin in nano-capsules [27]. This effect could be explained by the limited bioavailability of curcumin attributed to its molecular structure. However, it has been demonstrated that nano-formulated curcumin exhibits higher efficacy and faster cellular absorption than free curcumin [103]. Curcumin plays a significant role in glucose homeostasis that contributes to its potential benefits in diabetes management [103]. In this context, curcumin participates in several mechanisms, including activating glycolysis, inhibiting gluconeogenesis, and reducing hepatic lipid metabolism. Moreover, curcumin enhances insulin sensitivity by mitigating insulin resistance and by promoting pancreatic β cell function through its anti-inflammatory and antioxidant properties via NF-KB (nuclear factor kappa-light-chain-enhancer of activated B cells) suppression [104,105]. Additionally, curcumin lowers fasting glucose levels, according to our meta-analysis. Supplementation with turmeric achieves reductions of around 12 mg/dL by enhancing the activity of PPAR-γ (Peroxisome Proliferator-Activated Receptor γ), stimulating insulin secretion from the pancreas, and enhancing glucose uptake by upregulating the gene expression of glucose transporters. Moreover, it suppresses glucose production in the liver by enhancing AMP kinase activation and inhibiting glucose 6 phosphate kinase [106,107,108].

Out of the ten studies that evaluated the impact of ginger supplementation on glycemic metabolism, six showed a significant decrease in fasting glucose levels [29,30,31,49,52,53], while only five showed a significant decrease in HbA1c [29,30,31,35,52]. Four of them showed a significant decrease in insulin values after ginger supplementation [31,35,52,59]. The differences between the analyzed studies could be due to variations in the chemical composition of the administered ginger extract, the method of preparation, the type of ginger rhizome used, or differences in storage time [109,110]. However, most of selected articles did not explain the source of ginger used for the protocol of supplementation, and the dosages varied from 1.2 g/day [49] to 3 g/day [30,31,32], as well as the duration of the studies. In addition, one of them administered the supplementation of ginger in combination with black tea [32] and another study in combination with metformin [52]. Another factor that could explain the discrepancies among studies may be attributed to variations in individual responses. These variations could be linked to differences in patient characteristics at the start of the research, encompassing factors like the initial condition of the experimental group, body weight, the degree of insulin resistance, and other measured variables [30]. In this meta-analysis, the evaluated studies included newly T2DM diagnosed subjects [52], subjects with a T2DM diagnosis of more than two years ago [29] or ten years ago [30], or even T2DM subjects with end-stage renal disease who were on hemodialysis [53]. Several studies have proposed that ginger’s hypoglycemic effects can be attributed to its content of phenols, polyphenols, and flavonoids [111]. Our meta-analysis showed that ginger is the unique spice that reported a significant reduction in fasting glucose, HbA1c, and insulin levels after supplementation. In fact, ginger supplementation achieved a significant decrease in HbA1c similar to iSGLT2, Sitagliptin, and Vildagliptin drugs [112,113]. Ginger appears to mitigate insulin resistance by promoting the translocation of GLUT4 from the cytosol to the cell membrane [114]. Another potential impact of a ginger hydroalcoholic extract is the inhibition of hepatic glycogen phosphorylase enzyme, thereby preventing glycogen breakdown in the liver. Furthermore, ginger inhibits hepatic glucose phosphatase enzyme activity while increasing the activity of enzymes engaged in glycogen synthesis [115]. In this sense, Isa et al. suggested that the glucose-regulating and insulin-sensitizing effects of ginger could be due to PPAR-γ agonistic activity and/or the upregulation of adiponectin [116].

Eight studies evaluated the effect of black cumin supplementation on the glycemic profile, and all of them found significant differences in the fasting glucose after supplementation [37,46,47,54,55,60,64]. However, only five of these studies provided HbA1c data, and all exhibited a significant reduction in HbA1c levels [37,46,47,54,64]. Moreover, only four articles included insulin values pre- and post-supplementation, and two reported a significant decrease [54,60]. Black cumin, also known as Nigella or kalonji, is a species frequently found in Iran, scientifically referred to as Nigella sativa [117]. The meta-analysis conducted by Mahmoodi et al. [118] elucidated that the efficacy of Nigella sativa preparations depends on factors such as the dosage forms, the active ingredients prescribed, and the duration of the intervention. This study concluded that the most efficient approach to supplementing Nigella sativa for improving glycemic parameters involves a daily dosage of 2 g of its powdered form for a minimum of 12 weeks. However, in our meta-analysis, the dosage concentration in the studies varied from 1 g [37] to 3 g [47], and the dosage forms included Nigella sativa capsules made from crushed seeds [60,64], soft gel capsules containing Nigella sativa oil [37,38,47,54], or Nigella Sativa mineral oil [46,55]. In addition, the duration of the different studies was 56 days [37,38,60,64] or 84 days [46,47,54,55]. As discussed earlier, according to the results obtained in our meta-analysis, it appears that the administration protocol of Nigella sativa does not significantly influence glycemic parameters. Different studies have reported several mechanisms of action of the antidiabetic properties of Nigella sativa, such as an in vitro/in vivo inhibitory effect on pancreatic α-amylase and α-glucosidase, decreasing oxidative stress, and preserving pancreatic β-cell integrity and intestinal glucose absorption. The main bioactive compound of Nigella sativa is thymoquinone, and it has been shown to reduce hepatic glucose production and serum glucose levels, as well as insulin, mediating its effect through the activation of the insulin and AMP-activated protein kinase (AMPK) pathways [119].

Out of the eleven studies which analyzed the effect of saffron supplementation on the glycemic profile, six reported a significant decrease in fasting glucose [33,41,42,44,45,48], while only four showed a significant reduction in HbA1c [39,40,42,48] or insulin values [33,39,45,61]. These discrepancies could be due to the features of the T2DM population included in each study, intervention time, prescribed saffron dosage, or the fact that it was combined with physical exercise. Our meta-analysis reported that saffron achieves a significant reduction only in fasting glucose; it is also the herb that produces the smallest drop in fasting glucose. Similar results were reported by the meta-analysis performed by Giannoulaki et al. [120], concluding that the saffron supplementation achieved a significant reduction only in fasting glucose, including in T2DM or metabolic syndrome subjects with no discrimination among diseases. However, another meta-analysis, conducted by Correia et al. [121], showed that saffron supplementation significantly reduces fasting glucose, HbA1c, and postprandial blood glucose. Nonetheless, in this meta-analysis, all types of subjects are included, regardless of their associated pathologies. Saffron contains volatile components, such as safranal, and non-volatile components, which are carotenoids such as crocin, picrocrocin, and two vitamins, riboflavin and thiamine [122]. The mechanism of action of saffron in reducing the carbohydrate profile has been studied in many in vivo and in vitro studies [123,124,125,126]. These carotenoids have been shown to increase insulin sensitivity, improve pancreatic beta cell function, enhance the production and activity of antioxidant enzymes, and decrease oxidative stress indices and inflammation markers such as TNF-alpha [123,124,126,127]. Another study has suggested that saffron consumption and exercise could improve diabetic parameters through redox-mediated mechanisms and the GLUT4/AMPK pathway to trap glucose uptake [125]. In addition, saffron has been shown to exhibit antioxidant, neuroprotective, anti-inflammatory, antidepressant, and cardiovascular effects [123].

Our study has some limitations that are worth commenting on. Firstly, although cinnamon, turmeric, ginger, black cumin, and saffron have shown a significant decrease in fasting glucose according to our meta-analysis results, different factors can affect fasting glucose levels, such as changes in body weight or body mass index and the combination of spice or aromatic herb supplementation with physical activity or lifestyle changes. And all these factors have not been taken into account in most studies. Secondly, there is a wide heterogeneity in the quality of the studies, which partly limits the results that could be obtained in this meta-analysis. In general, few studies perform adequate statistics or even take into account changes in anthropometric characteristics in these statistical analyses, to evaluate whether the improvement in the carbohydrate profile can be attenuated or exacerbated by these environmental factors. Thirdly, although our meta-analysis shows the decreases in fasting glucose, HbA1c, or insulin that occurred with each type of herb consumed, it has not been possible to consider the effective dosage of supplementation prescribed for each herb due to the heterogeneous dosage observed between studies. This review emphasizes the potential therapeutic benefits of these spices in managing diabetes; however, additional research is needed to establish the most effective dosage and the availability of their active components. This is crucial for their practical use in treatment.

## 5. Conclusions

In conclusion, this is a large systematic review, with 77 studies included, and meta-analysis, with 45 studies included, that has evaluated the effect of all aromatic herbs and spices included in the MedDiet, such as black cumin, clove, parsley, saffron, thyme, ginger, black pepper, rosemary, turmeric, basil, oregano, and cinnamon, on the glycemic profile of individuals with T2DM. Our results showed that cinnamon, turmeric, ginger, black cumin, and saffron significantly decreased fasting glucose in T2DM subjects. Black cumin achieved the greatest decrease in the fasting glucose, followed by cinnamon and ginger. However, only ginger and black cumin reported a significant improvement in HbA1c, and only cinnamon and ginger showed a significant decrease in the insulin concentration. Of note, ginger appears to be the unique one out of the analyzed aromatic herbs in the MedDiet producing a significant decrease in the three outcomes examined, fasting glucose, HbA1c, and insulin. Finally, more studies are necessary to analyze the effect of clove, parsley, thyme, black pepper, rosemary, basil, and oregano on the glycemic profile in T2DM subjects.

## Figures and Tables

**Figure 1 nutrients-16-00756-f001:**
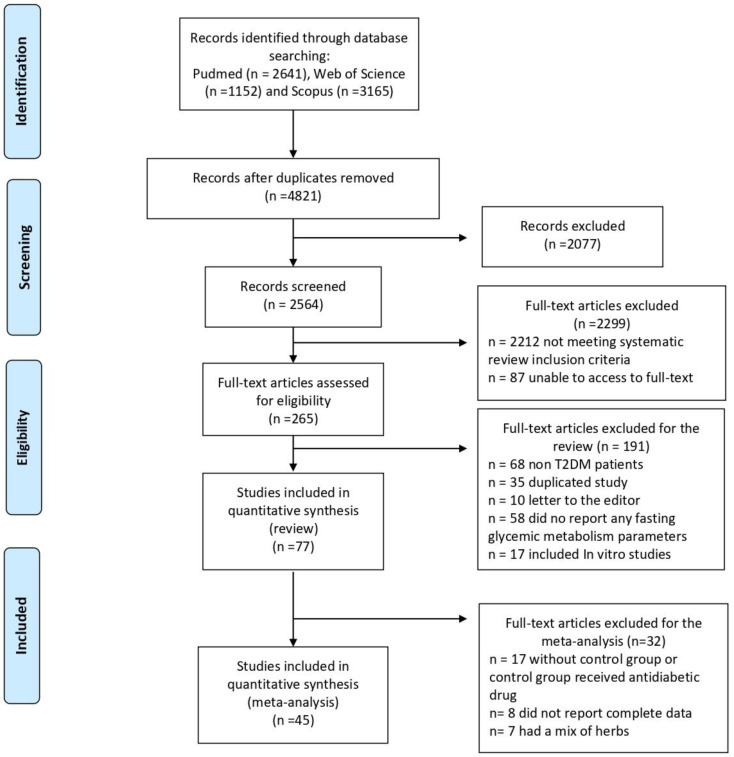
Flow chart.

**Figure 2 nutrients-16-00756-f002:**
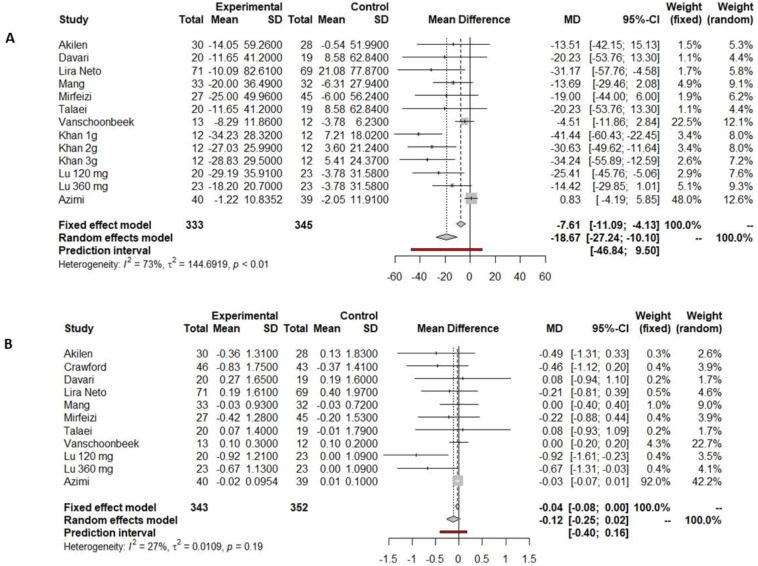
Forest plot showing the effects of cinnamon on fasting glucose (**A**) and HbA1c (**B**). 
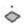
 Indicated results of fixed effect model, 
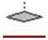
 indicated results of random effects models and 
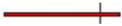
 indicated prediction interval of predictive value.

**Figure 3 nutrients-16-00756-f003:**
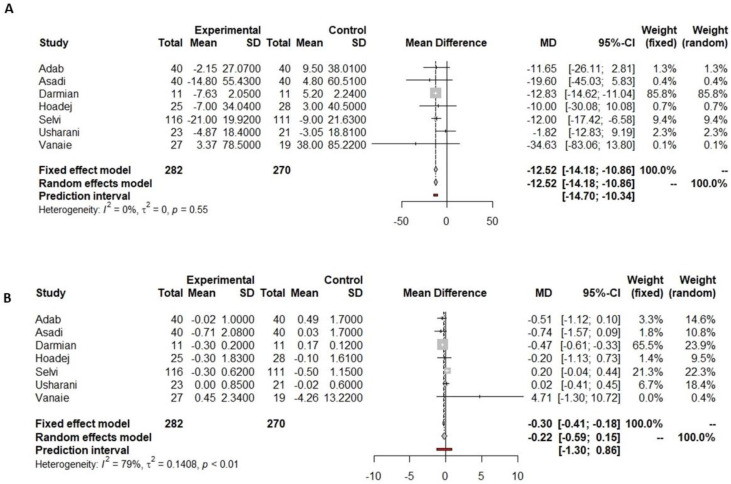
Forest plot showing the effects of curcumin on fasting glucose (**A**) and HbA1c (**B**). 
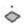
 Indicated results of fixed effect model, 
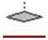
 indicated results of random effects models and 
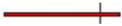
 indicated prediction interval of predictive value.

**Figure 4 nutrients-16-00756-f004:**
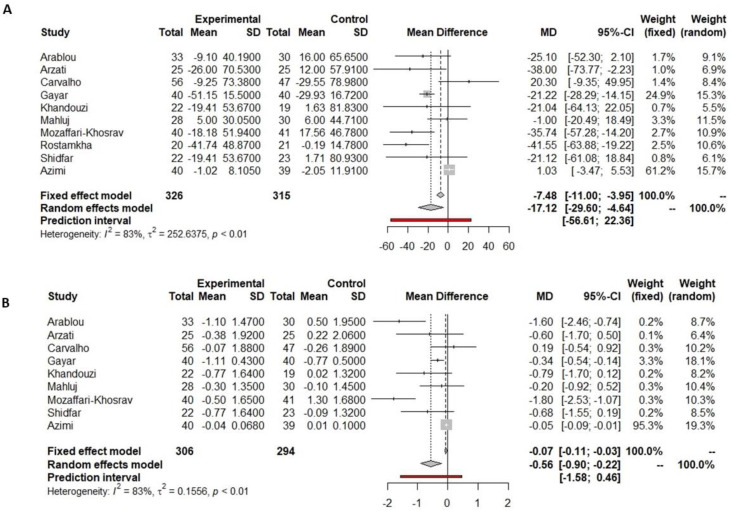
Forest plot showing the effects of ginger on fasting glucose (**A**) and HbA1c (**B**). 
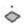
 Indicated results of fixed effect model, 
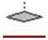
 indicated results of random effects models and 
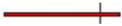
 indicated prediction interval of predictive value.

**Figure 5 nutrients-16-00756-f005:**
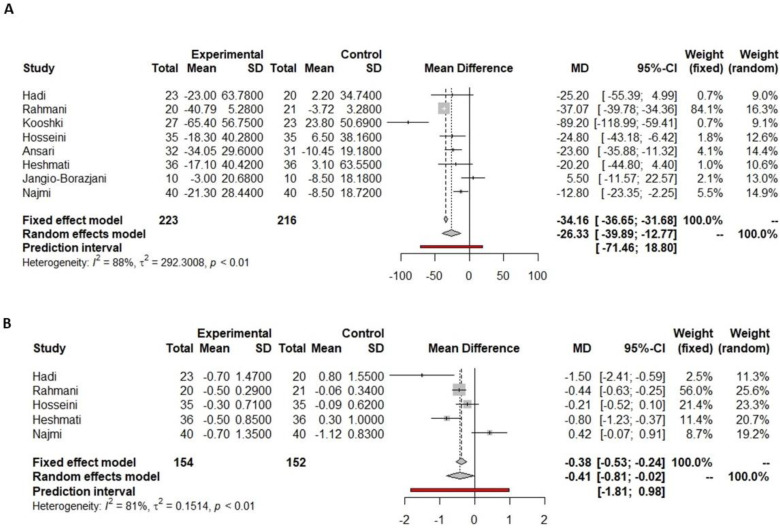
Forest plot showing the effects of black cumin on fasting glucose (**A**) and HbA1c (**B**). 
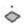
 Indicated results of fixed effect model, 
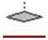
 indicated results of random effects models and 
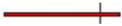
 indicated prediction interval of predictive value.

**Figure 6 nutrients-16-00756-f006:**
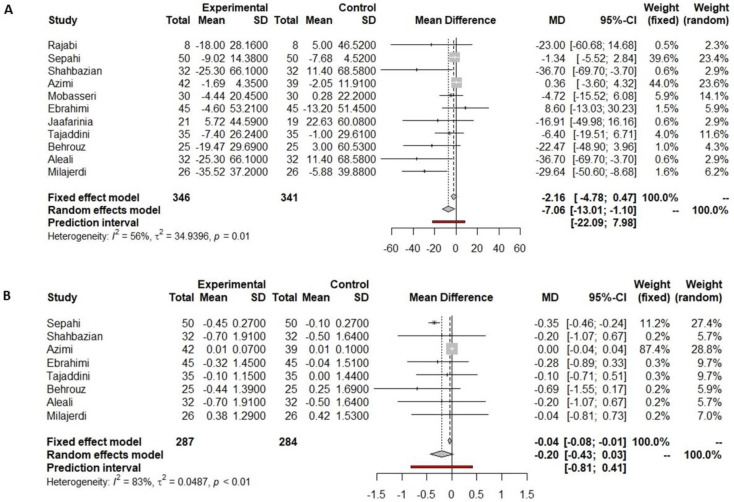
Forest plot showing the effects of saffron on fasting glucose (**A**) and HbA1c (**B**). 
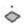
 Indicated results of fixed effect model, 
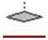
 indicated results of random effects models and 
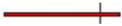
 indicated prediction interval of predictive value.

**Figure 7 nutrients-16-00756-f007:**
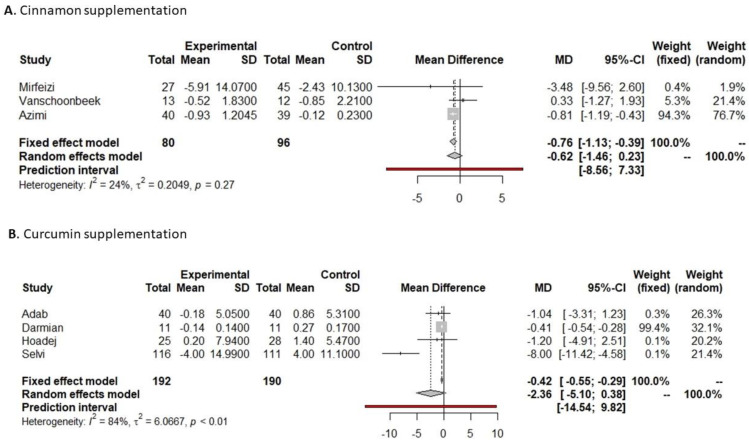
Forest plot of insulin after cinnamon or curcumin supplementation. 
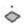
 Indicated results of fixed effect model, 
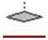
 indicated results of random effects models and 
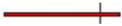
 indicated prediction interval of predictive value.

**Figure 8 nutrients-16-00756-f008:**
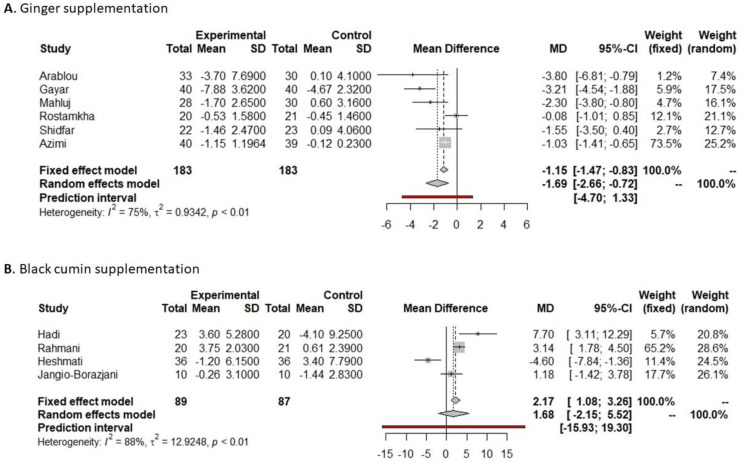
Forest plot of insulin after ginger or black cumin supplementation. 
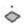
 Indicated results of fixed effect model, 
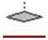
 indicated results of random effects models and 
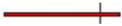
 indicated prediction interval of predictive value.

**Figure 9 nutrients-16-00756-f009:**
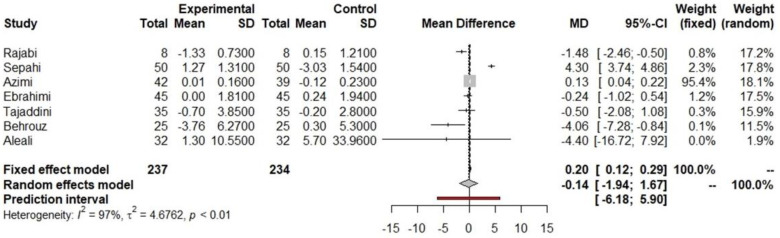
Forest plot of insulin after saffron supplementation. 
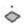
 Indicated results of fixed effect model, 
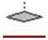
 indicated results of random effects models and 
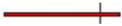
 indicated prediction interval of predictive value.

**Table 1 nutrients-16-00756-t001:** Summary of articles included in the quantitative synthesis (meta-analysis).

First Author, Year of Publication	N	Participants	Male, *n* (%)	Age, Years	Study Design	Intervention Dosage mg/day (Number of Subjects)	Duration (Days)	Body Weight (kg)		Glucose (mg/dL)		HbA1c (%)		Insulin (UI/µL)		Quality Checklist Mean
								Baseline	Final	Baseline	Final	Baseline	Final	Baseline	Final	
Akilen et al., 2010 [43]	58	T2D subjects treated with oral hypoglycemic agents, 18 years of age or older. Patients treated with insulin therapy, those with chronic disease, and pregnant or lactating women were excluded.	11 (36.6)	54.90 ± 10.14	Prospective, randomized, placebo-controlled, double-blind clinical trial. These patients were randomly assigned to placebo (n = 28) or cinnamon (n = 30) groups.	Cinnamon group (N = 30): received cinnamon capsules (500 mg) per day	84 days	87.6 ± 17.5	84.7 ± 16.4	159 ± 62.2	145 ± 55.9	8.22 ± 1.16	7.86 ± 1.42 *	NR	NR	0.785
			15 (53.6)	54.43 ± 12.53		Placebo group (N = 28): received placebo capsules (500 mg) per day		87.52 ± 20.24	87.02 ± 18.88	158 ± 46.7	157 ± 56.0	8.55 ± 1.82	8.68 ± 1.83	NR	NR	
Davari et al., 2020 [50]	39	Newly diagnosed T2D subjects, age 25–75 years, BMI 18–30 kg/m^2^, and T2D-diagnosed for less than 8 years. Pregnancy or patients with chronic disease were excluded.	8 (40%)		Randomized, double-blind, placebo-controlled clinical trial. All patients were randomized into two groups: cinnamon and control group.	Cinnamon group (N = 20): received three capsules of 1 g cinnamon extract (3 g of cinnamon per day)	56 days	73.75 ± 10.74	NR	183.85 ± 36.16	172.20 ± 44.86	10.04 ± 1.30	10.31 ± 1.86	9.85 (7.92–19.22)	12.10 (10.65–18.45)	0.661
			7 (36.8%)			Control group (N = 19): received three capsules of microcrystalline cellulose		77.15 ± 15.63	NR	190.57 ± 70.58	199.15 ± 49.86	10.11 ± 1.49	10.30 ± 1.70	10.60 (8.80–17.30)	12.20 (9.30–14.20)	
Lira Neto et al., 2022 [22]	140	T2D non-insulin subjects, age 18–80 years, and HbA1c > 6.0%. Patients with chronic disease, pregnancy, or allergic reaction to cinnamon were excluded.	51 (71.8)	61.7 (11.7)	Randomized, triple-blind, placebo-controlled clinical trial. All patients were randomized into two groups: cinnamon and control.	Cinnamon group (N = 71): received 3 g/day of cinnamon in capsules	90 days	NR	NR	10.3 (4.59)	9.77 (4.58) *	8.5 (2.3)	8.3 (2.2)	−0.01 (−12.20, 7.20)		0.857
			46 (66.7)	60.8 (10.8)		Control group (N = 69): received placebo; capsules were identical in both groups		NR	NR	9.00 (3.84)	10.17 (4.68)	8.0 (1.8)	8.4 (2.1)	−0.40 (−7.20, 11.30)		
Mang et al., 2006 [23]	79	T2D non-insulin treatment.	21 (63.6)	62.8 ± 8.37	Randomized, placebo-controlled, double-blind design study. All patients were randomized into two groups: cinnamon and placebo.	Cinnamon group (N = 33): received 1 g of cinnamon per day in capsules	121 days	NR	NR	9.26 ± 2.26	8.15 ± 1.65 *	6.86 ± 1.00	6.83 ± 0.83	NR	NR	0.411
			23 (71.9)	63.7 ± 7.17		Placebo group (N = 32): received placebo capsules (microcrystalline cellulose)		NR	NR	8.66 ± 1.47	8.31 ± 1.62	6.71 ± 0.73	6.68 ± 0.70	NR	NR	
Mirfeizi et al., 2016 [24]	105	T2D non-insulin-therapy subjects, with FBS > 140 mg/dL and HbA1c > 7%. Patients with chronic disease or with specific dietary needs or pregnancy were excluded.	3 (11.1)	52 ± 13	Multicenter stratified randomization (triple-blind) placebo-controlled. All patients were randomized into three parallel groups: cinnamon, Caucasian whortleberry, and placebo.	Cinnamon group (N = 27): received 1000 mg per day of cinnamon in capsules	84 days	28.4 ± 3.27	27.8 ± 3.01 *	180 ± 56	155 ± 40 *	8.52 ± 1.32	8.10 ± 1.24 *	21.6 ± 15.7	15.7 ± 11.4 *	0.786
			9 (30)	55 ± 10		Caucasian whortleberry (N = 30): received 1000 mg/day of whortleberry		28.6 ± 3.27	28.3 ± 3.69	199 ± 79	154 ± 39 *	8.80 ± 1.60	8.20 ± 1.41 *	22.5 ± 24.2	12.7 ± 8.68 *	
			11 (24.4)	54 ± 12		Placebo group (N = 45): received 1000 mg/day of starch capsules		28.9 ± 4.45	28.8 ± 4.33	172 ± 53	166 ± 59	8.58 ± 1.38	8.38 ± 1.65	20.0 ± 11.1	17.6 ± 8.67	
Talaei et al., 2017 [25]	39	T2D non-insulin-therapy subjects, FBS: <180 mg/dL, and T2D history < 8 years. Pregnancy, consumption of specific medicines, or chronic disease were excluded.	8 (40)	58.90 ± 7.93	Double-blind, randomized, placebo-controlled clinical trial. All patients were randomized into two groups: placebo and intervention.	Intervention group (N = 20): received three capsules of 1 g of cinnamon per capsule (3 g of cinnamon/day)	56 days	73.8 ± 10.7	NR	184 ± 36.2	172 ± 44.9	10.0 ± 1.30	10.1 ± 1.49	9.85 (7.92–19.2)	12.10 (10.7–18.5)	0.512
			7 (36.8)	56.26 ± 9.46		Placebo group (N = 19): received three capsules with microcrystalline cellulose as placebo per day		77.2 ± 15.6	NR	191 ± 70.6	199 ± 49.9	10.3 ± 1.86	10.3 ± 1.70	10.6 (8.80–17.3)	12.2 (9.30–14.2)	
Vanschoonbeek et al., 2006 [62]	25	Postmenopausal T2D women, non-insulin-dependent, and with stable medication for last 3 months.	0 (0)	64 ± 2	Double-blind, placebo-controlled trial. All patients were randomized into two groups: placebo and cinnamon.	Placebo group (N = 13): received 1500 mg/d placebo (wheat flour)	42 days	NR	NR	149 ± 5.95	145 ± 6.49	7.1 ± 0.2	7.2 ± 0.2	15.5 ± 2.16	14.62 ± 2.25	0.444
			0 (0)	62 ± 2		Cinnamon group (N = 12): received 1500 mg/d of cinnamon capsules (Cinnamomum cassia)		NR	NR	151 ± 10.6	143 ± 12.8	7.4 ± 0.3	7.5 ± 0.3	15.3 ± 1.81	14.8 ± 1.84	
Khan et al., 2003 [26]	60	T2D non-insulin subjects, age > 40 years, and FBS 140–400 mg/dL. Patients who were taking other medicine for other health conditions were excluded.	30 (50)	52.0 ± 6.87	Randomized clinical trial. All participants were divided into six groups: three received different gr of cinnamon, while another three-groups received placebo.	Group 1 (N = 12): received 1 g of cinnamon capsule per day	40 days	NR	NR	209 ± 30.6	175 ± 25.2	NR	NR	NR	NR	0.356
						Group 2 (N = 12): received 2 g of cinnamon capsule per day		NR	NR	205 21.6	178 ± 28.8	NR	NR	NR	NR	
						Group 3 (N = 12): received 3 g of cinnamon capsule per day		NR	NR	234 ± 25.2	205 ± 32.4	NR	NR	NR	NR	
				52.0 ± 5.85		Group 4 (N = 12): received 1 capsule of placebo		NR	NR	220 ± 18.0	227 ± 18.0	NR	NR	NR	NR	
						Group 5 (N = 12): received 2 capsules of placebo		NR	NR	223 ± 18.0	227 ± 23.4	NR	NR	NR	NR	
						Group 6 (N = 12): received 3 capsules of placebo		NR	NR	301 ± 25.2	306 ± 23.4	NR	NR	NR	NR	
Lu et al., 2012 [56]	66	T2D subjects with HbA1c > 7% and FBS > 8.0 mmol/L.	8(40)	62.4 ± 7.9	Randomized, double-blinded clinical study. All participants were randomly divided into 3 groups: placebo, low-dosage, and high-dosage groups. All patients were taking gliclazide (30 mg/day).	Low-dosage group (N = 20): received 120 mg of cinnamon capsule per day	84 days	NR	NR	11.2 ± 2.21	9.59 ± 1.66 *	8.92 ± 1.35	8.00 ± 1.00 *	NR	NR	0.511
			8 (34.8)	58.9 ± 6.4		High-dosage group (N = 23): received 360 mg of cinnamon capsule per day		NR	NR	9.00 ± 1.23	7.99 ± 1.05 *	8.90 ± 1.24	8.23 ± 0.99 *	NR	NR	
			9 (39.1)	60 ± 5.9		Placebo group (N = 23): received placebo capsules		NR	NR	8.92 ± 1.21	8.71 ± 2.01	8.93 ± 1.14	8.93 ± 1.04	NR	NR	
Crawford et al., 2009 [57]	89	T2D subjects with HbA1c > 7%. Pregnancy, age < 18 years, and allergy to cinnamon were exclusion criteria.	32 (58)	60.5 ± 10.7	Randomized clinical trial. Enrolled subjects were randomized into two groups: cinnamon (C. cassia) and control group.	Cinnamon group (N = 46): received capsules (500 mg each) of Cinnamomum cassia; they were instructed to take 2 capsules daily	90 days	31.9 ± 6.4	NR	NR	NR	8.47 ± 1.8	7.64 ± 1.7 *	NR	NR	0.536
			32 (59)	59.9 ± 9.2		Control group (N = 43): did not receive any supplementation		32.9 ± 6.4	NR	NR	NR	8.28 ± 1.3	7.91 ± 1.5	NR	NR	
Adab et al., 2019 [63]	80	Hyperlipidemic T2D patients, FBS < 200 mg/dL, HbA1C > 6%, TG > 150 mg/dL, or LDL-c > 100 mg/dL, BMI: 20–35 kg/m^2^, no insulin therapy, and no use of polyphenols or multivitamin supplements.	19 (48.7)	54.76 ± 6.00	Randomized, double-blind clinical trial. Eligible patients were randomly divided into two groups: the intervention (n = 40) and placebo (n = 40) groups.	Intervention group: received 2100 mg turmeric powder (three 700 mg turmeric capsules after main meals)	56 days	76.9 ± 10.4	75.1 ± 9.96 *	134 ± 25.6	132 ± 28.33	7.06 ± 1.01	7.04 ± 0.98	7.29 ± 4.92	7.11 ± 5.17	0.911
			17 (47.2)	55.66 ± 8.64		Placebo group: received 2100 mg corn starch flour as placebo (three 700 mg capsules after main meals)		74.6 ± 17.0	76.7 ± 14.4	130 ± 33.0	139 ± 41.6	6.79 ± 1.08	7.28 ± 1.59 *	7.29 ± 4.77	8.15 ± 5.72	
Asadi et al., 2019 [27]	80	T2D not insulin-dependent patients, aged 30–60 years, and BMI 25 to 39.9 kg/m^2^. Patients with chronic disease, pregnancy, or lactating were excluded.	5 (12.5)	53.3 (6.5)	Double-blind randomized, parallel, placebo-controlled clinical trial study conducted using intervention and placebo groups.	Intervention group (N = 40): received 80 mg of nano-curcumin capsules	56 days	77.4 (10.9)	77.1 (10.9)	166 (52.3)	151 (58.1) *	8.89 (2.18)	8.18 (1.96) *	NR	NR	0.856
			5 (12.5)	54.6 (6.2)		Placebo group: received 80 mg of polysorbate		75.9 (12.4)	75.9 (12.2)	185 (58.3)	190 (62.5)	9.19 (1.68)	9.22 (1.72)	NR	NR	
Darmian et al., 2021 [34]	42	T2D non-insulin-dependent (type II) diabetes, HbA1C > 6, Triglycerides (TG) > 150 mg/dL, LDL > 100 mg/dL, and BMI = 25–30 kg/m^2^.	NR	43.02 ± 3.04	Single-blind, randomized, placebo-controlled study. Subjects were randomly assigned to four groups, namely AT + TS, AT + placebo, TS, and control + placebo. The participants in the AT group were required to exercise at home three times per week. Each training session included 20 min at 60% of HRmax, 40 min at 75% of HRmax, and a 10 min cool-down. HRmax was calculated as = 220 – age.	Group AT+ TS (N = 11): received 2100 mg capsules containing turmeric powder daily	56 days	73.1 ± 2.91	69.2 ± 3.22 *	153 ± 1.75	135 ± 2.36 *	7.68 ± 0.48	6.93 ± 0.64 *	6.69 ± 0.13	5.98 ± 0.19 *	0.786
			NR	42.13 ± 2.39		Group AT+ placebo (N = 11): received 2100 mg capsules containing cornstarch flour daily		75.1 ± 2.07	72.2 ± 1.01 *	155 ± 1.48	142 ± 2.11 *	7.93 ± 0.69	7.06 ± 0.45 *	6.59 ± 0.08	6.28 ± 0.05 *	
			NR	44.33 ± 1.23		Group TS (N = 11): received 2100 mg capsules containing turmeric powder daily		74.1 ± 2.68	72.2 ± 1.76 *	155 ± 2.04	147 ± 2.06 *	7.70 ± 0.22	7.40 ± 0.16 *	6.55 ± 0.16	6.41 ± 0.06 *	
			NR	44.22 ± 3.07		Group control + placebo (N = 11): received 2100 mg capsules containing cornstarch flour daily		75.1 ± 3.20	78.4 ± 4.21 *	153 ± 2.50	159 ± 1.84 *	7.75 ± 0.13	7.92 ± 0.11 *	6.63 ± 0.18	6.90 ± 0.13 *	
Hodaei et al., 2019 [28]	53	T2D not insulin-dependent patients, aged 40–70 years old, and BMI 18.5–35 kg/m^2^. Patients with chronic disease and multivitamin supplements were excluded.	15 (61.6)	58 ± 8	Randomized, double-blind, placebo-controlled trial. All patients were randomized into two groups: curcumin group and placebo. All patients were followed-up by phone every 15 days.	Curcumin group (n = 25) received three capsules of 500 mg of curcumin; 21 subjects of this group completed the trial	70 days	78 ± 13.28	77 ± 13.6 *	160 ± 35	153 ± 33 *	11.3 ± 1.6	11 ± 2	9.2 ± 9	9.4 ± 6	0.878
			11 (39.1)	60 ± 7		Placebo group (n = 28) received three capsules of placebo (444 mg of cooked rice flour); 23 subjects of this group completed the trial		74.04 ± 11.5	74.23 ± 12.3	144 ± 40.6	147 ± 40.4	11.2 ± 1.3	11.1 ± 1.8	8.3 ± 6	9.7 ± 4.7	
Selvi et al., 2013 [51]	60	T2D subjects with T2D diagnosed < 2 years.	30 (100)	46.8 ± 6.1	Open-label randomized clinical trial. All T2D patients were randomized into two groups: one treatment only with metformin and another with metformin + turmeric.	Group 1: T2D subjects’ treatment with metformin (500 mg) twice a day	28 days	24.1 ± 3.26 kg/m^2 †^	NR	111 ± 24	102 ± 18 *	7.8 ± 0.5	7.5 ± 0.7	23 ± 16.4	19 ± 13	0.515
			30 (100)	47 ± 7.17		Group 2: T2D subjects’ treatment with metformin (500 mg) twice a day + turmeric capsules (2 g/day).		23.4 ± 3.03 kg/m^2 †^	NR	116 ± 23	95 ± 11.4 *	7.9 ± 1.3	7.4 ± 0.9 *	18 ± 9.9	22 ± 12	
Usharani et al., 2008 [65]	72	T2D subjects aged 21–80 years and taking stable T2D medications for 2 months. Uncontrolled T2D, smoking, or patients with other chronic diseases were excluded.	11 (47.8)	55.52 ± 10.76	Randomized, parallel-group, placebo-controlled trial. Subjects were randomized into NCB-02 (new formula with curcumin), atorvastatin, or placebo.	NCB-02 group (N = 23): received new formulation with curcumin, demethoxy curcumin, and bisdemethoxy; this capsule contained curcumin 150 mg; they received it twice per day	56 days	63.6 ± 10.7	NR	155 ± 17.9	150 ± 18.8	8.04 ± 0.85	8.04 ± 0.85	NR	NR	0.452
			12 (52.2)	50.47 ± 10.35		Atorvastatin (N = 23): received 10 mg of atorvastatin daily		64.6 ± 9.27	NR	161 ± 19.7	158 ± 16.5	8.30 ± 0.86	8.29 ± 0.81	NR	NR	
			11 (52.4)	49.75 ± 8.18		Placebo (N = 21): two capsules daily		61.5 ± 8.63	NR	161 ± 20.0	158 ± 17.4	7.82 ± 0.57	7.80 ± 0.62	NR	NR	
Vanaie et al., 2019 [58]	46	T2D patients on oral antidiabetic drugs or insulin, age ≥ 18 years, overt proteinuria, eGFR ≥ 30 mL/min/1.73 m^2^, and controlled blood pressure.	16 [59%]	59 ± 6.25	Randomized, double-blind, controlled trial. Patients were randomized into two groups (curcumin and placebo).	Curcumin group (N = 27): the patients received 500 mg curcumin capsule three times/day after meal (1500 mg/day)	56 days	NR	NR	184 ± 75.4	187 ± 81.3	9.46 ± 2.25	9.91 ± 2.42	NR	NR	0.570
			11 [58%]	61 ± 10.80		Placebo group (N = 19): the patients received a placebo capsule with a similar packing		NR	NR	176 ± 73.0	214 ± 93.6	13.0 ± 14.17	8.75 ± 2.17	NR	NR	
Arablou et al., 2014 [35]	70	T2D non-insulin-dependent subjects, HbA1C 7–10%, BMI 20–35 kg/m^2^, no pregnancy, no use of tobacco or alcohol, and no chronic disease.	8 (24.2)	52.6 ± 8.4	Double-blinded, placebo-controlled clinical trial. Participants allocated randomly into two groups receiving ginger or placebo.	Ginger group (N = 33): received two capsules per day, which contained 1600 mg of ginger	84 days	66.2 ± 8.2	66.1 ± 8.2	131 ± 42.5	122 ± 37.4	8.4 ± 1.6	7.3 ± 1.3 *	8.3 ± 8.3	4.6 ± 1.4 *	0.714
			7 (23.3)	52.0 ± 9.0		Control group (N = 30): received placebo capsules (containing wheat flour)		66.1 ± 7.8	66.0 ± 7.7	129 ± 62.5	145 ± 68.4	8.1 ± 1.5	8.6 ± 2.2	6.9 ± 4.6	7.0 ± 3.3	
Arzati et al., 2017 [36]	50	T2D not insulin-dependent patients, BMI 18.5–35 kg/m^2^, and age 30–60 years.	9 (34.8)	51.7 ± 8.5	Double-blind placebo-controlled trial study. All T2D subjects were randomly allocated to 2 groups of intervention and placebo.	Intervention group (N = 25): received 2000 mg per day of ginger capsules	70 days	78.4 ± 11.7	77.9 ± 11.2	170 ± 74.8	144 ± 65.3	7.30 ± 1.90	6.92 ± 1.93	NR	NR	0.676
			7 (27.3)	49.6 ± 8.6		Control group (N = 25): received 2000 mg per day of placebo supplements		76.7 ± 14.2	76.7 ± 14.0	161 ± 49.0	173 ± 63.9	7.50 ± 2.03	7.72 ± 2.08	NR	NR	
Carvalho et al., 2020 [49]	103	T2D subjects, with HbA1c 6–10%, with oral hypoglycemic agents.	31 (30.1%)	58.64 ± 11.11	Double-blind, parallel, randomized control trial. All patients were divided into two groups: control and intervention.	Control group (N = 56): received 600 mg per day of cellulose supplement in capsules	84 days	NR	NR	185 ± 74.2	176 ± 72.6 *	8.36 ± 1.89	8.29 ± 1.86	NR	NR	0.832
						Intervention group (N = 47): received 600 mg per day of ginger supplement		NR	NR	204 ± 88.2	174 ± 64.1 *	8.40 ± 1.96	8.14 ± 1.81	NR	NR	
El Gayar et al., 2019 [52]	80	T2DM newly diagnosed subjects, HbA1c < 9%, and BMI ≥ 30 kg/m^2^. Pregnancy and patients with chronic disease were excluded.	19 (47.5)	46.35 ± 9.53	A randomized, single-blind, placebo-controlled clinical trial. Subjects were randomly divided into two groups: ginger and placebo groups. All patients had to maintain a diet and constant PA.	Ginger group (N = 40): consumed three capsules daily, each capsule containing 600-mg of ginger powder (total daily dosage was 1.8 g) + 1000 mg of metformin	56 days	32.4 ± 1.51 kg/m^2 †^	31.8 ± 1.21 * kg/m^2 †^	172 ± 17.9	121 ± 9.06 *	8.05 ± 0.46	6.94 ± 0.38 *	20.7 ± 4.14 mIU/L	12.9 ± 2.59 * mIU/L	0.748
			22 (55)	46.10 ± 8.66		Placebo group (N = 40): received three placebo capsules (wheat flour) + 1000 mg of metformin		32.3 ± 1.39 kg/m^2 †^	32.3 ± 1.39 kg/m^2 †^	182 ± 18.8	152 ± 13.2 *	8.03 ± 0.54	7.26 ± 0.45 *	17.9 ± 2.50	13.2 ± 2.08 *	
Khandouzi et al., 2015 [29]	41	T2D non-insulin therapy patients, aged 20–60 years, with T2D diagnosis for more than 2 years. Patients with chronic disease were excluded.	5 (22.7)	45.20 ± 7.64	Randomized, double-blind, placebo-controlled clinical trial. Patients were divided randomly into two groups: experimental and control.	Experimental group (N = 22): received 2 g/day of ginger powder supplement in capsules	84 days	No significant differences in BMI at the beginning and the end of the study in both groups		162 ± 58.0	142 ± 47.9 *	7.37 ± 1.86	6.60 ± 1.26 *	NR	NR	0.643
			9 (47.4)	47.10 ± 8.31		Control group (N = 19): received 2 g/day of lactose supplement, as placebo				155 ± 81.8	157 ± 81.8 *	7.30 ± 1.31	7.32 ± 1.32	NR	NR	
Mahluj et al., 2013 [59]	64	T2D subjects with normal blood pressure, aged 38–65 years, and mean BMI 29.5 kg/m^2^.	14 (43.8)	49.2 ± 5.1	Randomized, double-blind, placebo-controlled trial. All participants were randomized into two groups: intervention and placebo.	Intervention group (N = 28 completed study): received one tablet of ginger twice a day (2 g/day) immediately after lunch and dinner	56 days	79.3 ± 11.8	79.1 ± 11.4	142 ± 34	147 ± 23	7.0 ± 1.3	6.7 ± 1.4	12.7 ± 2.9	11.0 ± 2.3 *	0.714
			16 (50)	53.1 ± 7.9		Placebo group (N = 30 patients completed study): received one tablet of placebo twice a day		76.8 ± 14.5	76.9 ± 14.1	153 ± 47	159 ± 42	6.9 ± 1.4	6.8 ± 1.5	11.5 ± 3.0	12.1 ± 3.3	
Mozaffari-Khosravi et al., 2014 [30]	88	T2D non-insulin subjects for at least 10 years, FBS < 180, no pregnancy or lactation, no autoimmune or chronic disease, BMI < 40 kg/m^2^, and no consumption of lipid-lowering drugs.	13 (32.5)	49.83 ± 7.23	Randomized, double-blind, placebo-controlled trial. The patients were categorized into 2 groups of ginger (GG) and placebo (PG).	Ginger group (N = 40): consumed daily 3 one-gram capsules containing ginger powder, after taking meals	56 days	28.1 ± 5.29 kg/m^2 †^	28.1 ± 5.33 kg/m^2 †^	171 ± 54.91	153 ± 48.34 *	8.2 ± 1.6	7.7 ± 1.7 *	NR	NR	0.732
			18 (43.9)	51.05 ± 7.70		Placebo group (N = 41): consumed daily 3 cellulose microcrystalline capsules, after taking meals		28.51 ± 4.95 kg/m^2 †^	28.53 ± 0.03 kg/m^2 †^	136 ± 40.53	154 ± 50.57	6.9 ± 1.3	8.2 ± 1.9 *	NR	NR	
Rostamkhani et al., 2023 [53]	41	T2D subjects with end-stage renal disease who were on hemodialysis, aged > 18 years, free of any acute gastrointestinal issues, thyroid abnormalities, gallstones, or a history of ginger sensitivity.	11 (50%)	60.05 ± 11.12	Randomized, double-blind, controlled parallel-group study. The participants were allocated into intervention and control groups.	Intervention group (N = 20): received four capsules with 500 mg of ginger per day (2000 mg of ginger powder daily)	56 days	69.7 ± 10.8	69.8 ± 10.4	175 ± 56.1	133 ± 33.2 *	NR	NR	11.2 ± 1.68	10.6 ± 1.47	0.818
			12 (54.5%)	59.64 ± 10.69		Control group (N = 21): received four placebo capsules containing starch		74.6 ± 14.3	74.4 ± 15.2	150 ± 34.0	157 ± 34.5	NR	NR	10.5 ± 1.54	10.1 ± 1.37	
Shidfar et al., 2015 [31]	45	T2D non-insulin and non-smoking subjects, age 20–60 years, BMI < 30 kg/m^2^, and HbA1c 6–8%. Patients with chronic disease, pregnancy, or multivitamin supplementation were excluded.	NR	45.2 ± 7.64	Double-blind, parallel, randomized clinical trial. The patients were stratified by sex and BMI and randomly assigned into two groups: ginger or placebo.	Ginger group (N = 22): received 3 g of powdered ginger capsules daily (each capsule contained 1 g)	84 days	81.2 ± 13.25	80.0 ± 13.2	162 ± 58	142 ± 47.9 *	7.37 ± 1.86	6.60 ± 1.26 *	5.97 ± 2.76	4.51 ± 2.01 *	0.712
			NR	47.1 ± 8.31		Placebo group (N = 23): received 3 g of daily placebo (lactose) capsules		78.5 ± 14.1	78.2 ± 13.4	155 ± 81.8	157 ± 81.8	7.39 ± 1.31	7.30 ± 1.32	6.43 ± 3.98	6.52 ± 4.14	
Hadi et al., 2021 [37]	43	T2D subjects with BMI of 25–35 kg/m^2^, aged 30–60 years, non-smokers, not currently receiving insulin therapy, and did not have history of other diseases.	10 (43.5)	51.4 ± 9.2	Double-blind randomized, controlled clinical trial was conducted among two groups (intervention and control) running in parallel.	Intervention group (N = 23): received two soft gel capsules containing 500 mg of Nigella sativa per day	56 days	28.4 ± 4.4 kg/m^2 †^	27.6 ± 4.09 * kg/m^2 †^	190 ± 71.5	167 ± 51.0 *	7.9 ± 1.6	7.2 ± 1.3 *	8.2 ± 3.2	11.8 ± 6.1	0.723
			10 (50)	56.00 ± 3.4		Control group (N = 20): received daily two soft gel capsules containing oil or sunflower oil		28.8 ± 8.1 kg/m^2 †^	29.6 ± 7.7 kg/m^2 †^	154 ± 35.7	156 ± 33.7	7.7 ± 1.5	8.5 ± 1.6	16.6 ± 10.6	12.5 ± 6.4	
Rahmani et al., 2022 [54]	41	T2D hemodialysis subjects aged 20 to 60 years, BMI 18.5 to 30 kg/m^2^, three HD sessions per week, six months on HD, and willingness to participate in the study. Exclusion criteria were pregnancy or lactation and cigarette smoking, among others.	12 (60.0)	49.60 (8.75)	Randomized, double-blinded, placebo-controlled, parallel-group clinical trial. Patients were divided into two groups: Nigella sativa group (NS) or placebo group using random allocation software. All patients were requested not to change their PA and diet during the study.	Nigella sativa group (N = 20): received two g/d of NS oil soft gel capsules (one capsule, twice daily)	84 days	79.2 ± 12.55	NR	190.70 ± 6.08	149.91 (2.68) *	8.26 ± 0.33)	7.76 ± 0.23 *	15.9 ± 2.07	19.7 ± 1.98 *	0.761
			11 (52.4)	48.57 (10.5)		Placebo group (N = 21): received the same amount of paraffin oil; both NS oil and paraffin oil capsules were packaged in dark containers with similar colors, smells, and appearances; each container included 30 capsules		78.4 ± 10.99	NR	157 ± 3.43	153 ± 3.10	8.38 ± 0.37)	8.32 ± 0.31)	19.4 ± 2.49	20.0 ± 2.28	
Kooshki et al., 2019 [38]	50	T2D patients aged 35–64 years old and BMI of 25–34 kg/m^2^. Subjects with infection diseases, renal or thyroid diseases, hepatitis, cancer, or stroke; those on cholesterol-lowering drugs or insulin were excluded.	7 (25.9)	52.30 (9.43)	Randomized, double-blind clinical trial study. Patients were divided into two groups: intervention or placebo. Subjects were advised not to change their dietary habits, PA, and drug regimens. The 24 h food recall and PA questionnaires were evaluated.	Intervention group (N = 27): received 1000 mg N. sativa oil as two capsules, each containing 500 mg N. sativa oil, daily	56 days	29.01 (3.48) kg/m^2 †^	NR	219 ± 64	153.6 ± 44.2 *	NR	NR	NR	NR	0.747
			9 (39.1)	55.91 (8.98)		Placebo group (N = 23): received two placebo capsules containing medium-chain triglyceride oils at lunch and dinner		28.1 (4.45) kg/m^2 †^	NR	173 ± 47.2	196 ± 53.3	NR	NR	NR	NR	
Hosseini et al., 2013 [46]	70	T2D patients with FBG 140–180 mg/dL, body weight 55–75 kg, age 34–63 years, taking no more than 500 mg metformin.	14 (40)	48.74 ± 7.33	Randomized double-blind study. Patients were divided into two groups: N. sativa and placebo group.	N. sativa group (N = 35): received 5 mL daily N. sativa oil	84 days	30.8 (3.55) kg/m^2 †^	29.52 (3.50) * kg/m^2 †^	180 ± 31.8	162 ± 45.3 *	8.82 ± 0.73	8.52 ± 0.68 *	NR	NR	0.464
			16 (46)	50.72 ± 5.69		Placebo group (N = 35): received 5 mL daily mineral oil (placebo)		30.92 (3.67) kg/m^2 †^	31.12 (3.73) kg/m^2 †^	180 ± 32.3	186 ± 42.1	8.79 ± 0.55	8.70 ± 0.67	NR	NR	
Ansari et al., 2017 [55]	63	T2D subjects with CKD (Stage 3 and 4) due to diabetic nephropathy aged 20–60 years were included. Pregnant females, patients on dialysis, terminally sick, immune-deficient, or having severe renal pathology were excluded.	NR	48.09	Prospective, randomized, parallel-group, and open-label study. T2D patients were randomized into two groups: control and intervention.	Control group (N = 31): received conservative management (insulin, torsemide, telmisartan, iron, calcium, Vitamin D3, and erythropoietin) of diabetic nephropathy	84 days	NR	NR	138 ± 33.1	104 ± 9.30 *	NR	NR	NR	NR	0.416
			NR	53.27		Intervention group (N = 32) received conservative management along with N. sativa oil (2.5 mL, per orally, once daily)		NR	NR	114 ± 22.0	104 ± 13.2 *	NR	NR	NR	NR	
Heshmati et al., 2015 [47]	80	T2D patients aged 30–60 years old, T2D diagnosed for more than six months and taking antidiabetic medications. Patients with CVD, renal, hepatic, or pancreatic diseases were excluded.	16 (45.7)	45.3 ± 6.5	Double-blind, placebo-controlled, randomized clinical trial. Patients were randomly divided into two groups: the intervention group received Nigella sativa oil soft gel capsules, and the control group received the placebo oil.	Intervention group (N = 36): received 3 g/day Nigella sativa oil soft gel capsules (one three times a day)	84 days	77.7 ± 11.4	74.8 ± 11.3 *	183 ± 42.1	166 ± 38.5 *	8.3 ± 0.9	7.8 ± 0.8 *	12.2 ± 7.1 mg/dL	11.0 ± 3.3 mg/dL	0.947
			17 (48.6)	47.5 ± 8.0		Control group (N = 36): received sunflower oil as placebo; both NS oil and sunflower capsules were provided for subjects in similar opaque bottles		76.6 ± 13.7	77.3 ± 14.0	202 ± 63.9	205 ± 63.2	8.3 ± 1.0	8.6 ± 1.0 *	10.3 ± 9.0 mg/dL	13.7 ± 4.6 mg/dL	
Jangjo-Borazjani et al., 2023 [60]	40	T2D middle-aged women without previous CVD. The exclusion criteria included previous or current insulin therapy, history of cardiovascular disease, conditions that would preclude physical activity, and use of antioxidant, anti-inflammatory, and corticosteroid medicines.	0 (0)	43.23 ± 3.45	Randomized, double-blind clinical trial. Subjects were randomly assigned to 4 groups: resistance training + Nigella sativa (RN), Nigella sativa (NS), resistance training + placebo (RP), and control group (CO). Subjects of the RN and RP groups performed resistance training 3 days per week. Each session comprised a 10 min warm-up, 45 min resistance training, and a 10 min cool-down.	RN group (training + Nigella supplementation) (N = 10): received four N. sativa capsules (500 ± 10 mg), taking 2 g of N. sativa per day	56 days	76.3 ± 12.58	66.0 ± 4.59	142 ± 21.1	117 ± 12.3 *	NR	NR	11.0 ± 4.19	5.76 ± 2.48 *	0.607
				44.2 ± 4		NS group (Nigella supplementation) (N = 10): received four N. sativa capsules (500 ± 10 mg), taking 2 g of N. sativa per day		66.6 ± 6.61	66.77 ± 6.08	132.40 ± 23.63	129.40 ± 14.81 *	NR	NR	10.23 ± 3.53	9.97 ± 2.25 *	
				44.13 ± 1.19		RP group (training + placebo) (N = 10): received four capsules with maltodextrin (500 ± 10 mg) as a placebo per day		74.5 ± 12.75	72.99 ± 6.67	118.30 ± 17.45	119.3 ± 8.43	NR	NR	6.92 ± 2.95	7.40 ± 1.37	
				42.9 ± 3.2		Control group (N = 10): received four capsules with maltodextrin (500 ± 10 mg) as a placebo per day		70.64 ± 7.02	69.34 ± 4.98	150.70 ± 19.20	142.20 ± 16.94	NR	NR	11.55 ± 2.91	10.11 ± 2.75	
Najmi et al., 2012 [64]	80	Newly detected patients of metabolic syndrome with T2D (HbA1C > 7%), aged 20–70 years. The exclusion criteria were pregnancy, T1D, CVD, impaired liver function test, chronic renal disease, or familial dyslipidemia.	52 (65)	20–70 years	Open-label randomized controlled study. Patients were randomly divided into two groups (n = 40 each). In group I (Std group), patients received metformin and atorvastatin. In group II (NSO group), patients received Nigella sativa as add-on therapy.	Std group (N = 40): received metformin 500 mg twice a day and atorvastatin 10 mg once a day	56 days	NR	NR	165.6 ± 32.6	144.3 ± 12.9 *	8.11 ± 0.83	6.99 ± 0.83 *	NR	NR	0.381
						NSO group (N = 40): received 500 mg capsule of Nigella sativa as add-on therapy; aspirin 150 mg once a day was given in both group		NR	NR	144.2 ± 21.6	135.7 ± 11.6	7.71 ±0.73	7.18 ± 0.70	NR	NR	
Rajabi et al., 2022 [61]	32	Obese women with T2DM without CVD and musculoskeletal disorders, HbA1c < 9.9%, no diabetic complications, no regular AT, no smoking, DM history less of than 5 years, and a maximum of one type of oral antidiabetic tablet a day.	0 (0)	51.5 ± 6.16	Participants were divided into four groups: saffron + training (ST) (n = 8).	Powdered saffron (400 mg) was placed in capsules and used for two months	56 days	81.0 ± 5.01	77.6 ± 6.37	185 ± 30	128 ± 32 **	NR	NR	7.72 ± 1.92	5.00 ± 1.25 ^1^	0.536
				57.62 ± 6.81	Placebo + training (PT) (n = 8).	Placebo capsules containing 400 mg of wheat flour and used for two months		81.9 ± 3.30	80.1 ± 3.47	194 ± 42	175 ± 3 **	NR	NR	8.51 ± 1.45	6.75 ± 0.95 **	
				54.12 ± 7.37	Saffron supplementation (SS) (n = 8).	Powdered saffron (400 mg) was placed in capsules and used for two months		81.5 ± 6.91	79.6 ± 7.47	190 ± 31	172 ± 7	NR	NR	8.13 ± 0.75	6.80 ± 0.70 **	
				56.87 ± 5.11	Placebo (P) (n = 8).	Placebo capsules containing 400 mg of wheat flour and used for two months		87.0 ± 5.90	87.2 ± 6.32	215 ± 42	220 ± 50	NR	NR	8.95 ± 1.10	9.10 ± 1.30	
Sepahi et al., 2022 [39]	150	Patients with DM2 who did not use insulin, not well-controlled diabetes mellitus, age > 18, and HbA1c > 7. Patients with CKD and/or hepatic failure and mothers during pregnancy or lactating periods were excluded from the study.	22 (44)	57.58 ± 1.0	Placebo-controlled triple-blinded clinical trial, where DM2 participants were divided into three groups: 50 subjects received saffron.	The saffron tablets contained 15 mg saffron. Crocin, placebo, and saffron tablets were prepared in a similar shape, color, and size, stored in a dark container, and coded by a pharmacist	84 days	NR	NR	171 ± 9.41	162 ± 16.6	7.92 ± 0.2	7.47 ± 0.31 *	10.9 ± 0.94	12.2 ± 1.5	0.795
			21 (42)	57.16 ± 1.5	50 subjects received crocin.	The crocin tablets contained 15 mg crocin		NR	NR	185 ± 12.1	164 ± 14.4 *	8 ± 0.22	7.46 ± 0.25 *	11.5 ± 1.13	10.8 ± 1.34	
			25 (50)	56.92 ± 1.9	50 subjects received placebo.	The placebo tablets contained 15 mg placebo		NR	NR	161 ± 4.33	154 ± 4.69 *	7.84 ± 0.23	7.74 ± 0.3	14.4 ± 1.60	11.4 ± 1.48 *	
Shahbazian et al., 2019 [42]	64	T2DM patients aged 30–65 years old, using oral hypoglycemic agents, having FBS ≥ 126 mg/dL and an HbA1c ≥ 7%. The exclusion criteria included pregnancy or lactating, chronic T2DM complications, or insulin treatment, among others.	11 (34.4)	52.4 ± 13	Randomized double-blind clinical trial. All T2D patients included were randomized into two groups: saffron and control group. A 24 h dietary recall questionnaire was completed. The patients were asked not to change their diet, medication, and physical activity.	Control group (N = 32) received two placebo capsules per day; these placebo capsules contained lactose, magnesium stearate, and starch	84 days	27.5 ± 4.2 kg/m^2 †^	NR	177 ± 60.1	189 ± 74.7	8.80 ± 1.8	8.3 ± 1.4	NR	NR	0.818
						Saffron group (N = 32) received two capsules (each 15 mg saffron) per day (30 mg/day)		28.8 ± 4.0 kg/m^2 †^	NR	173 ± 73.9	148 ± 53.5 *	8.9 ± 2.0	8.2 ± 1.8 *	NR	NR	
Azimi et al., 2014 [32]	204	Subjects with T2D (FBS ≥ 126 mg/dL), aged ≥30 years, BMI ≥ 25 kg/m^2^, not on insulin therapy, and not taking medications except metformin or glibenclamide. Exclusion criteria included pregnancy, starting insulin therapy, or consumption of cinnamon, cardamom, ginger, or saffron during the running period.	16 (0.40)	54.15 ± 1.0	Parallel, randomized, single-blind, placebo- controlled clinical trial. Before intervention, all participants were included in a three-week run-in period to match their tea consumption. The patients were randomly assigned to four intervention groups, cardamom, cinnamon, ginger, and saffron, and one control group.	Cinnamon group (N = 40) received 3 g cinnamon in three glasses of black tea	56 days	75.6 ± 1.20	75.3 ± 1.20	359 ± 10.8	358 ± 10.9 *	7.89 ± 0.10	7.87 ± 0.09	11.4 ± 0.17	11.3 ± 0.17	0.714
			17 (40.5)	51.59 ± 1.3		Cardamon group (N = 42) received 3 g cardamom in three glasses of black tea		78.6 ± 1.20	78.5 ± 1.20	361 ± 12.3	359 ± 12.04	7.89 ± 0.10	7.87 ± 0.10	11.2 ± 0.20	11.2 ± 0.19	
			16 (0.38)	57.02 ± 1.0		Saffron group (N = 42) received 1 g saffron in three glasses of black tea		82.0 ± 1.0	81.9 ± 0.99	358 ± 4.30	357 ± 4.39	7.73 ± 0.07	7.74 ± 0.07	11.0 ± 0.15	11.0 ± 0.15	
			15 (0.37)	55.21 ± 1.1		Ginger group (N = 41) received 3 g ginger in three glasses of black tea		79.4 ± 0.9	79.2 ± 0.96	367 ± 8.09	366 ± 8.12	7.94 ± 0.069	7.90 ± 0.067	11.8 ± 0.15	11.6 ± 0.18	
			15 (0.38)	53.64 ± 1.3		Control group (N = 39): received 1 g placebo in three glasses of black tea		78.7 ± 1.2	78.5 ± 1.1	355 ± 11.9	353 ± 12.0	7.50 ± 0.10	7.51 ± 0.10	11.0 ± 0.22	10.9 ± 0.22	
Mobasseri et al., 2020 [44]	60	T2D subjects with FBS> 126 mg/dL, HbA1c > 6.5%, with BMI 25 to 35 kg/m^2^, and having T2DM for at least six months and using antidiabetic drugs. Exclusion criteria were using insulin and hormone replacement therapy and using any antioxidant supplements, among others.	NR	50.57 ± 9.88	Randomized, double-blind, placebo-controlled clinical Trial (allocation ratio 1:1) was carried out with 60 T2D patients. These 60 patients were randomly allocated to one of the two treatment groups: saffron group (n = 30) and placebo group (n = 30). All the patients were asked to keep their dietary intake or PA as usual.	Saffron group (N = 30) received 100 mg/day saffron capsules (1 capsule) per day	56 days	83.0 ± 11.47		135 ± 19.6	131 ± 21.2 *	NR	NR	NR	NR	0.818
			NR	51.63 ± 11.30		Control group (N = 30) received starch capsules (1 capsule) per day		85.4 ± 14.2		135 ± 21.3	135 ± 23.0	NR	NR	NR	NR	
Ebrahimi et al., 2019 [40]	80	T2D subjects, aged 30–70 years, HbA1c 6.5–10%, taking no nutritional supplements, no smoking, alcohol abuse, and BMI 20–35 kg/m^2^. The exclusion criteria included insulin therapy and changes in drug treatment or PA.	20 (50)	55.2 ± 7.3	Prospective, double-blind, placebo-controlled, randomized study. Subjects were randomly allocated to the saffron supplement group (n = 45) or placebo group (n = 45).	Saffron group received daily a tablet containing 100 mg saffron twice a day	84 days	75.4 ± 12.8	74.2 ± 12.9 *	167 ± 53.7	162 ± 52.7	8.01 ± 1.40	7.69 ± 1.49 *	4.70 ± 1.7 pmol/L	4.70 ± 1.9 pmol/L	0.773
			16 (38)	53 ± 10.6		Control group received daily the same amount of placebo (maltodextrin)		80.3 ± 12.9	78.8 ± 18.1	161 ± 51.1	148 ± 51.8	7.38 ± 1.53	7.34 ± 1.48	4.47 ± 1.8 pmol/L	4.71 ± 2.05 pmol/L	
Jaafarinia et al., 2022 [66]	40	T2D patients aged ≥ 18 years, 5 years of history of T2DM, HbA1c < 8%, SBP < 160 or DBP < 100 mmHg, SCr levels ≤ 2 mg/dL, oral hypoglycemic or insulin treatment, or hypercholesterolemia within a statin. Exclusion criteria included eGFR < 30 mL/min/1.73 m^2^, CVD, alcohol dependency, or cigarette smoking among, others.	11 (57.90)	62.68 ± 9.84	Randomized, triple-blind, placebo-controlled, 2-arm, parallel-group, phase 2 clinical trial using a 1:1 ratio of allocation. Saffron group included 22 subjects, while 22 subjects were included in placebo group. Three patients, one from the saffron group and two from the placebo group, dropped out of the intervention study.	Saffron group (N = 21): patients received one tablet of crocin 15 mg daily	90 days	27.2 ± 3.86 kg/m^2 †^	27.0 ± 3.95 kg/m^2 †^	141 ± 36.7	146 ± 49.6	NR	NR	NR	NR	0.909
			12 (57.14)	63.86 ± 10.62		Placebo group (N = 19): patients received one tablet of crocin 15 mg daily		27.3 ± 3.34 kg/m^2 †^	27.2 ± 3.44 kg/m^2 †^	137 ± 57.2	159 ± 62.6 *	NR	NR	NR	NR	
Tajaddini et al., 2021 [33]	70	T2D subjects with BMI 25–35 kg/m^2^, aged 30–60 years. Insulin treatment, hormone replacement therapy and consumption of dietary or antioxidant supplements, history of surgery, serious illness, pregnancy, or lactation were excluded.	15 (50.0)	50.5 ± 9.8	Double-blind, randomized, placebo-controlled clinical trial. Seventy participants were randomly allocated to two groups: control (N = 35) and saffron group (N = 35). Both patients and assessors were blind to the allocation.	Saffron group (N = 35) received a capsule with 100 mg saffron powder per day, which should be taken daily before a meal	56 days	82.7 ± 11.3	82.4 ± 11.1	138 ± 21.6	131 ± 29.2 *	7.7 ± 1.2	7.6 ± 1.1	7.3 ± 3.8	6.6 ± 3.9 *	0.843
			13 (43.3)	51.8 ± 10.9		Control group (N = 35) received a capsule with 100 mg of maltodextrin per day, which should be taken daily before a meal		84.6 ± 14.4	84.3 ± 13.8	134 ± 29.2	133 ± 30.0	7.5 ± 1.6	7.5 ± 1.2	7.1 ± 2.7	6.9 ± 2.8	
Behrouz et al., 2020 [45]	50	T2D subjects, aged 30–70 years, BMI 18.5–30 kg/m^2^, and taking oral hypoglycemic agents. Insulin, herbal and/or nutritional supplements, glucocorticoids, and non-steroid anti-inflammatory drugs within 3 months, uncontrolled diabetes (HbA1c ≥ 8.5%), and patients with chronic diseases were excluded.	4 (16)	57.08 ± 7.41	Randomized, double-blind, single-center, parallel-group, controlled clinical trial. Patients were selected using a simple sampling procedure and stratified (1:1) into two groups randomly: crocin group (n = 25) or the placebo group (n = 25). Three subjects in saffron and two in control group dropped out of the study.	Saffron group: two tablets of 15 mg crocin were administered orally (15 mg/day)	84 days	77.1 ± 10.2	NR	149 ± 30.1	129 ± 29.31 *	7.80 ± 1.29	7.36 ± 1.47	17.3 ± 7.14 (mU/L)	13.5 ± 4.62 * (mU/L)	0.869
			3 (12)	59.86 ± 9.46		Control group: two tablets of 0 mg crocin were administered orally; placebo tablets were similar to the crocin supplements in terms of the size, color, shape, smell, and distribution bottles		74.18 ± 7.97	NR	157.18 ± 63.29	160.18 ± 57.34	7.61 ± 1.62	7.86 ± 1.75	15.0 ± 5.52 (mU/L)	15.3± 5.04 (mU/L)	
Aleali et al., 2019 [48]	64	T2D patients aged 30–65 years, taking oral hypoglycemic medicines and without diabetic complications. Pregnancy and breastfeeding, chronic complications of diabetes, insulin treatment, CVD history, smoking, alcohol intake, and anticoagulant therapy were excluded.	8 (25)	53.5 ± 9.9	Double-blind clinical trial. T2D patients were randomized into two groups: saffron group and control group. Saffron or placebo capsules were given for 2 weeks. Patients were followed by either telephone or face-to-face contact. Two 24 h food recall questionnaires were completed.	Saffron group (N = 32): received two capsules per day (in total, 30 mg saffron)	84 days	28.8 ± 4.0 kg/m^2 †^	NR	173 ± 73.9	148 ± 53.5 *	8.9 ± 2.0	8.2 ± 1.8 *	12.5 ± 9.9	13.8 ± 11.1	0.738
			11 (34.4)	52.4 ± 13		Control group (N = 32): received two placebo capsules that were identical to the main capsules		27.5 ± 4.2 kg/m^2 †^	NR	177 ± 60.1	189 ± 74.7	8.8 ± 1.8	8.3 ± 1.4	12.3 ± 8.2	18.0 ± 37.3	
Milajerdi et al., 2017 [41]	54	T2D subjects, aged 40–65 years, BMI 18.5–30 kg/m^2^. Smoking patients, insulin medications, uncontrolled blood glucose, high PA, pregnant, lactating, and those women who had planned for pregnancy were excluded.	6 (23.1)	54.57 ± 6.96	Randomized triple-blind clinical trial. Fifty-four T2D patients were randomized into two groups: saffron and control group. One person from the control and one from the saffron group left the study. Participants were asked not to change their diet, PA, or drugs during the intervention.	Saffron group (N = 26) received two capsules twice a day (in the morning and evening); each capsule contained 15 mg of saffron	56 days	63.1 ± 31.6	NR	164 ± 40.9	129 ± 31.9 *	6.37 ± 1.30	6.75 ± 1.28	NR	NR	0.839
			6 (23.1)	55.42 ± 7.58		Control group (N = 26) received two capsules twice a day (in the morning and evening); each capsule contained 15 mg of placebo		66.3 ± 9.01	NR	160 ± 38.4	154 ± 41.2	6.83 ± 1.36	7.25 ± 1.65	NR	NR	

* denotes a significant difference after intervention or supplementation; ** denotes a significant decrease compared to the control group; ^1^ indicates change from baseline. ^†^ indicates BMI due to body weight was not reported. NR: not reported. NS: non-significant difference before and after intervention. PA: physical activity. HD: hemodialysis; cfu: colony forming unit.

**Table 2 nutrients-16-00756-t002:** Evaluation of quality assessment instruments for randomized controlled trials included in the meta-analysis study.

Instruments	ObjectiveSufficiently Described	Study Design	Method of Subject	Comparison Group	Random Allocation	Blinding of Investigators	Blinding of Subjects	Outcome and Exposure Measure(s)	Sample Size	Analytic Methods	Estimate of Variance	Controlling for Confounding	Results Reported	Conclusion Supported
Akilen et al., 2010 [43]														
Davari et al., 2020 [50]														
Lira Neto et al., 2022 [22]														
Mang et al., 2006 [23]														
Mirfeizi et al., 2016 [24]														
Talei et al., 2017 [25]														
Vanschoobeek et al., 2006 [62]														
Khan et al., 2003 [26]														
Lu et al., 2012 [56]														
Crawford et al., 2009 [57]														
Adab et al., 2018 [63]														
Asadia et al., 2019 [27]														
Darmian et al., 2021 [34]														
Hodaei et al., 2019 [28]														
Selvi et al., 2013 [51]														
Usharani et al., 2008 [65]														
Vanaie et al., 2019 [58]														
Arablou et al., 2014 [35]														
Arzati et al., 2017 [36]														
Carvalho et al., 2020 [49]														
El Gayar et al., 2019 [52]														
Khandouzi et al., 2015 [29]														
Mahlujl et al., 2013 [59]														
Mozaffari-khosravi et al., 2014 [30]														
Rostamkhani et al., 2023 [53]														
Shidfar et al., 2015 [31]														
Hadi et al., 2020 [37]														
Rahmani et al., 2022 [54]														
Kooshki et al., 2019 [38]														
Hosseini et al., 2013 [46]														
Ansari et al., 2017 [55]														
Heshmati et al., 2015 [47]														
Jangjo-Borazjani et al., 2021 [60]														
Najmi et al., 2012 [64]														
Rajabi et al., 2022 [61]														
Sepahi et al., 2022 [39]														
Shahbazian et al., 2019 [42]														
Azimi et al., 2014 [32]						NA								
Mobasseri et al., 2020 [44]														
Ebrahimi et al., 2019 [40]														
Jaafarinia et al., 2022 [66]														
Tajaddini et al., 2021 [33]														
Behrouz et al., 2020 [45]														
Aleali et al., 2019 [48]														
Milajerdi et al., 2017 [41]						NA	NA							

Symbology significance and scoring is as follows: 

 Yes (2 points); 

 Partial (1 point); 

 No (0 points); NA denotes “Not applicable”. A complete description of the issues included in the quality assessment is as follows: (1) Question or objective sufficiently described? (2) Study design evident and appropriate? (3) Method of subject and comparison group selection or source of information and input variables described and appropriate? (4) Subject and comparison group (if applicable) characteristics sufficiently described? (5) If interventional and random allocation was possible, was it reported? (6) If interventional and blinding of investigators was possible, was it reported? (7) If interventional and blinding of subjects was possible, was it reported? (8) Outcome and (if applicable) exposure measure(s) well defined and robust to measurement, misclassification bias? Means of assessment reported? (9) Sample size appropriate? (10) Analytic methods described, justified, and appropriate? (11) Some estimate of variance is reported for the main results? (12) Controlling for confounding? (13) Results reported in sufficient detail? (14) Conclusion supported by the results?

## Data Availability

The data used to carry out this study will be provided upon request to the principal investigators.

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
