# Peer review of "Effect of Aromatic Herbs and Spices Present in the Mediterranean Diet on the Glycemic Profile in Type 2 Diabetes Subjects: A Systematic Review and Meta-Analysis"

_nutrients, 2024, doi:10.3390/nu16060756_

Round 1

Reviewer 1 Report

Comments and Suggestions for Authors

In the manuscript by Garza et al., the authors conduct a comprehensive analysis of the literature to elucidate the impact of Mediterranean Diet spices and herbs on glycemic regulation in individuals with type 2 diabetes. A total of 77 articles were evaluated, with 45 meeting the criteria for inclusion in a meta-analysis. The findings indicate that cinnamon, turmeric, ginger, black cumin, and saffron effectively reduce fasting glucose levels, with black cumin being the most potent, followed by cinnamon and ginger. Notably, ginger was the only spice to demonstrate reductions in fasting glucose, Hb1Ac, and insulin levels. However, the study's quality assessment revealed inconsistencies, particularly regarding blinding and analytical methods. 

  1. While the review underscores the therapeutic promise of these spices in diabetes care, further investigation is warranted to determine the optimal dosages and bioavailability of their active constituents, which are essential for practical application. 
  2. Many of the studies' intervention durations were brief, potentially obscuring the sustained effects of spice supplementation on glycemic control. 
  3. Additionally, the review's scope was limited to a subset of spices, as other herbs like clove, parsley, and thyme lacked sufficient data for analysis. 
  4. Attention also warrants being paid to the confounding influence of concurrent interventions, such as exercise or medication, on the spices' effects. 
  5. Lastly, the clinical significance of the statistical findings on glycemic parameters requires further clarification to enhance the research's translational value.

Reviewer 2 Report

Comments and Suggestions for Authors

This study aimed to assess the impact of aromatic herbs and spices commonly found in the Mediterranean Diet on the glycemic profile of individuals with T2DM. Through a systematic review of interventional studies retrieved from citation databases, the research synthesized data from 77 studies qualitatively and 45 studies quantitatively. The findings revealed significant effects of cinnamon, turmeric, ginger, black cumin, and saffron on glycemic control in T2DM subjects. Some issues need further consideration and improvement.

1.         It is advisable to ensure the consistency of the results reported by different divisions. For example, in the section discussing changes in blood glucose metabolism, the results of each herb should be presented in a similar format so that readers can compare them more easily.

2.         It is recommended to discuss the clinical significance of these spices or aromatic herbs in changing blood glucose parameters in the discussion section, rather than just statistical significance.

3.         The title of some sub-pictures is not marked, and it is recommended to properly describe the picture content in the image annotation.

Round 2

Reviewer 1 Report

Comments and Suggestions for Authors

For a clinical expert, factors like herbs, spices, and plant based medicinal approaches for therapies is bit of a Pandora's box especially due to lack of clinical models. However, it cannot be neglected that there are tons of manuscripts describing such positive effects. So, considering that, I feel that the authors have addressed all the comments satisfactorily. I encourage the authors to include a separate detailed section on the limitations of such studies.

Comments on the Quality of English Language

The article need thorough proofreading for grammatical errors including spelling, tenses, and punctuations.
